# An ultrafast plenoptic-camera system for high-resolution 3D particle tracking in unsegmented scintillators

Till Dieminger [1], Saúl Alonso-Monsalve [1], Christoph Alt[1], Claudio Bruschini [2], Noemi Bührer [1,3], Edoardo Charbon [2], Kodai Kaneyasu[2], Tim Weber[1], Matthew Franks[1] & Davide Sgalaberna [1] ✉

Neutrino detectors, particle calorimeters and some dark matter detectors require dense and massive active materials. An extremely fine segmentation is desirable to achieve precise three-dimensional particle tracking. However, such systems introduce significant challenges in construction and demand a large number of readout electronics channels, leading to extremely high costs. In this article, we propose an alternative approach to elementary particle detection that enables ultrafast three-dimensional high-resolution imaging in large volumes of unsegmented scintillator. Enabling technologies are plenoptic systems and time-resolving single-photon avalanche diode array imaging sensors. Together, they enabled us, using a plenoptic camera, to reconstruct the origin of single photons in the scintillator. A case study focused on neutrino detection demonstrates full event reconstruction with a spatial resolution of two hundred micrometres. This work paves the way for a class of particle detectors whose capabilities should be further enhanced through future developments and expanded to Cherenkov light detection, medical imaging and neutron detection.

Modern elementary particle detectors deployed at high-energy physics experiments often have to cover huge volumes or surfaces and, at the same time, provide a spatial resolution on the order of 100 μm with a time resolution of a few hundred picoseconds or better. Moreover, dense active materials are necessary for detecting weakly interacting particles, such as neutrinos[1–6] and certain categories of dark matter candidates[7–10], as well as for electromagnetic and hadronic calorimetry[11–15].

Scintillator-based detectors constitute a suitable choice for active, high-density detection systems, offering the potential for sub-millimetre spatial resolution. Organic scintillators emit blue or green isotropic light, typically in the range of 8000 to 13,000 photons per MeV loss, when traversed by ionising radiation[16], with a decay time down to about 1–2 ns[17,18]. Advances in detector technology have enabled 3D segmentation down to 1 cm with sub-nanosecond timing in

tonne-scale scintillator volumes with wavelength shifting (WLS) fibre readout[19–23]. R&D on 3D printing of plastic scintillator is also underway[24–30]. Another notable scintillator-based technology, currently under development, aims to get rid of the geometrical segmentation by deploying organic scintillating materials with a short optical scattering length to stochastically contain the visible photons within small localised volumes[31]. These detectors, typically read out by Silicon Photomultipliers (SiPMs)[32], are also suitable for precise time-of-flight measurements of both charged particles[33] and neutrons[34,35]. In the form of thin scintillating fibres, spatial resolutions below 100μm can be achieved[36,37]. However, the main drawback lies in the challenging and costly construction, due to the very large number of channels required, particularly when targeting tonne-scale detectors. However, the main drawback lies in the challenging and costly construction due to the very large number of analog readout channels required, each

[1]IPA, ETH Zürich, Zurich, Switzerland. [2]Advanced Quantum Architecture Lab (AQUA), EPFL, Neuchâtel, Switzerland. [3]Present address: University of Zürich, Zurich, Switzerland. ✉e-mail: davide.sgalaberna@cern.ch

needing to be instrumented with corresponding digitisation electronics, particularly when targeting tonne-scale detectors.

In recent years, a new class of photosensors has seen tremendous improvements: Single-Photon Avalanche Diode (SPAD) arrays. These arrays consist of multiple photosensitive diodes manufactured in CMOS technology, which are independently read out to provide single-photon images with sub-nanosecond time resolution. They are sometimes also referred to as digital SiPMs. Unlike SiPMs, SPAD arrays integrate readout electronics directly on-chip[38]. Consequently, a single data line allows the readout of millions of pixels, rather than requiring an independent analogue channel and digitisation chain for each. They are well-suited for applications including LIDAR[39], non-line-of-sight Imaging[40], Raman spectroscopy[41], and event cameras[42]. SPAD arrays featuring pixel pitches as small as $2.5\mu m$[43,44], or equipped with time-to-digital converters (TDCs) providing timestamp resolutions down to 50ps[39,45], have been demonstrated in the literature. Recently, fast CMOS SPAD array sensors have been proposed as a cost-effective light readout system for scintillating fibre-based particle detectors[46].

The next step in scintillator detectors involves 3D imaging of photon-starved particle events in monolithic volumes ref. [47] was able to reconstruct the position of gamma rays with a SPAD array sensor. However, the spatial resolution was limited by the use of the circle of confusion method, applicable only to point-like sources. Other independent works, based on Monte Carlo (MC) simulation studies, have proposed more complex optical systems adopting conventional photosensors to enhance the sensitivity to the spatial depth of particle interactions, showing potential resolutions around 1–10 cm, depending on the volume size: array of multiple lenses coupled to photomultiplier tubes (PMTs) for a large-scale unsegmented liquid scintillator detector[48]; coded masks[49] or objective lenses[50] coupled to SiPM to detect vacuum ultraviolet (VUV) scintillation light in liquid argon. In ref. [51], a system of three objective lenses viewing three orthogonal faces of a few-centimetre plastic scintillator volume for fast neutron detection was studied with MC simulations.

The device with the greatest potential for 3D imaging of photon-starved events is the plenoptic camera. It combines a single main objective lens (main lens) with a micro-lens array (MLA) placed in front of an imaging photosensor. Each micro-lens projected onto a subset of sensor pixels acts as a tiny camera with a slightly different perspective. This enables stereoscopic vision and thus depth reconstruction from a single exposure. A plenoptic camera is capable of capturing the "light field", described by the function $\mathcal{L}(x, y, z, \phi, \theta)$, which encodes the light intensity at each spatial point $(x, y, z)$ propagating in direction $(\phi, \theta)$[52]. For this reason, it is often referred to as a light-field camera. Although the concept dates back to 1908[53], it is only within the past two decades that the computationally intensive image post-processing has become practical[54–56]. Alternative plenoptic configurations have been proposed[57] and commercialised[58]. Recently, a plenoptic camera instrumented with a charge-coupled device (CCD) sensor has been proposed for 3D particle dosimetry[59]; however, it lacks the timing information and single-photon sensitivity required to resolve images of individual particles. Closely related to the concept of plenoptic cameras, refs. [60,61] investigates a design comprising lens arrays placed on the six faces of a 60 cm plastic scintillator cube, read out using SPAD array imagers, studied through MC simulations for gamma-ray detection via Compton-scattered electrons. As noted by the authors, a principal limitation of the approach is that tracks exceeding beyond the field of view of a single lens cannot be accurately reconstructed, thereby restricting the detector's capability to the tracking of particles below a certain energy threshold. A plenoptic system is mentioned as an alternative option, but was neither conceptualised nor studied. As of now, the application of plenoptic cameras with sub-nanosecond timing to image particle events remains unexplored.

In this article, we propose and demonstrate a paradigm shift in the detection of particles in a large unsegmented scintillating volume, resulting in high-resolution tracking and calorimetry of multi-particle events. The detector consists of a system of multiple plenoptic cameras, each one with an MLA behind the main lens and a SPAD array sensor with time-resolving capabilities. A pixel pitch of the order of 10 μm is desirable to constrain the light field properly. The post-processing of the 2D photon-starved images captured by the plenoptic cameras provides a pure 3D image by ray-tracing each detected photon back to its origin, leveraging the acquired light field information. If an organic scintillator with a decay time of a few nanoseconds is employed, a single-pixel timestamp resolution on the order of a few hundred picoseconds enables the effective suppression of the dominant fraction of dark counts via time-coincidence techniques. More generally, sub-nanosecond time resolution is often required in particle physics experiments to obtain a high-purity signal sample. The attenuation length, in both its scattering and absorption/emission components, shall be much longer than the size of the detector such that the original directions of the detected photons are preserved. This is easily achievable in organic scintillators with attenuation lengths of a few metres for plastics[17] and a few tens of metres for liquids[62,63]. In this work, we demonstrate a detector concept with a plenoptic camera instrumented with a SPAD array, and complement it with realistic MC simulation studies. We developed different methods for image post-processing and a data-driven calibration of the optical model. The first prototype of this detector concept, which we named PLATON-prototype (PLenoptic imAge of Tracked photONs), is shown in Fig. 1. Based on the results of the prototype measurement campaign, we provide the recipe of a future PLATON detector, whose development is underway, determining the design parameters of both the plenoptic system and the SPAD array sensor. We studied its design with an MC simulation of both a large cubic metre size detector as well as in a smaller PLATON-10cm module exposed to an accelerator neutrino beam. A deep neural network, based on the transformer architecture developed initially for large-language models[64,65], was developed to precisely capture both the lateral and depth spatial information with sub-millimetre resolution. This work lays the foundation for an entirely different approach to detecting elementary particles in dense materials, offering exceptionally high spatial and temporal resolution.

## Results

### The PLATON prototype

The PLATON-prototype, shown in Fig. 1, adopts SwissSPAD2[66] as the imaging sensor. It is a gated SPAD array with a 16.39 μm pixel pitch, a peak photon-detection efficiency of 5% at 520 nm (photon-detection probability of 50% at 520 nm and a fill factor of 10.5%, and no on-chip per-pixel micro-lenses) and a median dark count rate of 0.26 counts per second (cps) per $\mu m^2$ of active area. A single image frame corresponds to a gate window that can be varied between 10 ns and 100 μs. The SPAD array returns single-bit frames, where an active pixel indicates that its SPAD has been activated, either by a photon or thermal noise. 2D intensity images with greyscale information are obtained from a stack of multiple frames. The plenoptic system was made in collaboration with Raytrix GmbH[58], which designed and mounted the MLA (f/2.4, micro-lens diameter of 125 μm). During the assembly process, some glue leaked onto the MLA, making small areas on the left and right sides unusable for accurate imaging. This can be seen in the inset of Fig. 1. A faster MLA, such as f/1, desirable to maximise light collection efficiency, could not be used because the glob top, applied to protect the wire bonds of SwissSPAD2, was too thick to place the MLA sufficiently close to the sensor. This limitation will be overcome in a future prototype. A Voigtländer Nokton II 25 mm photographic lens, refitted with a C-Mount, was attached to the printed circuit board, on which the sensor is situated. The focus is set at a 30 cm distance to operate in a focused plenoptic camera mode, with a virtual image being formed behind the sensor. It was operated with f/2.4 to match the f-number of the MLA. With these specifications, we expect a depth

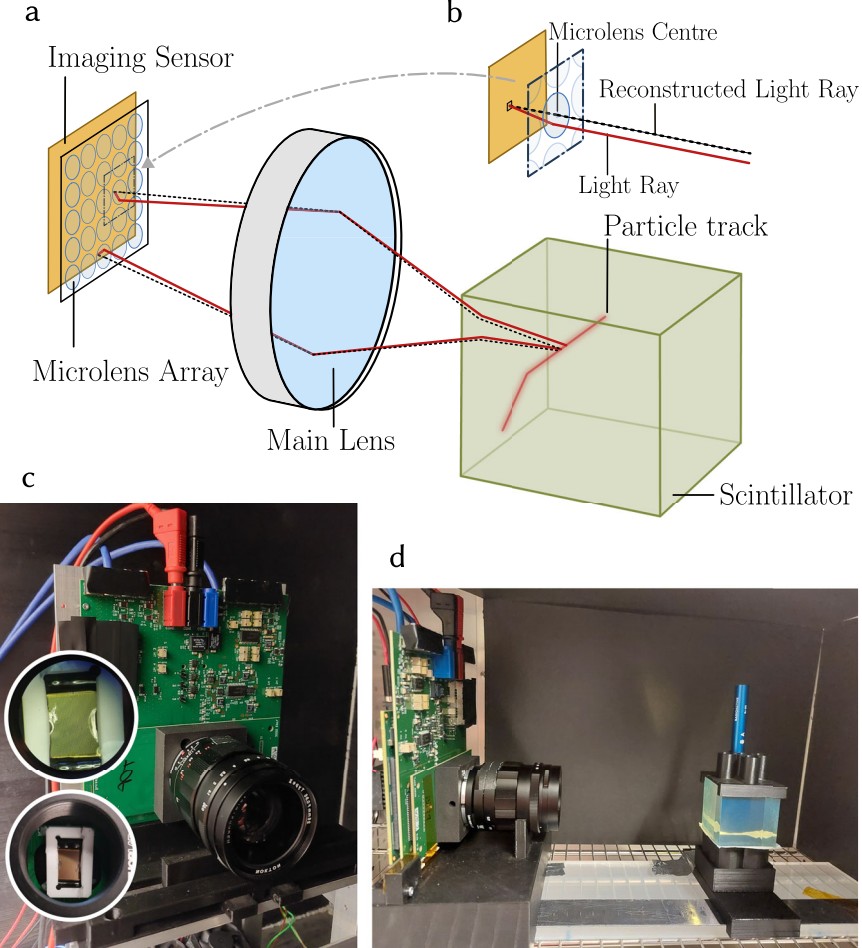

**Fig. 1 | PLATON detection and reconstruction principle and the PLATON camera prototype. a** The PLATON imaging principle. A plenoptic camera detects photons emitted by a charged particle traversing the scintillator. **b** Principle used to reconstruct single photon micro-images, using the micro-lens centre to determine the angle of the arriving photon. The intersection points of multiple photons are then used to reconstruct the particle track. **c** The PLATON camera prototype. Two insets show the ceramic MLA mount and a microscopy shot of the MLA itself. Here, the spill of the adhesive used to mount the MLA to the ceramic frame on the MLA can be seen on two sides. **d** The experimental setup, measuring ⁹⁰Sr-electrons using the PLATON-prototype.

resolution of the order of 1–5 mm and a sub-mm lateral resolution with a depth of field of 150 mm, calculated based on the effective resolution ratio (ERR) and the overall minimal change in virtual depth, as described in ref. 57. In the future, a better resolution will be achieved with a higher light collection efficiency and optics, currently constrained by the slower MLA f-number.

### 3D spatial resolution for a point-like light source

Using a back-illuminated pin-hole ($\phi$50 µm) attached to a motorised movement stage, two samples of data were collected. One was used for calibration, while the other was used for validating the post-processing method. To achieve this, the pinhole was positioned at 168 discrete locations relative to the camera using the motorised stage. These locations are illustrated in Fig. 2. Since the primary objective of this study was to calibrate the camera and evaluate the achievable spatial resolution, these measurements were performed without a scintillator, which would require a simple correction for the refractive index.

The image post-processing was performed as described in the "Image post-processing method" subsection of Methods. The calibration of the PLATON-prototype is described in the "Calibration of the PLATON-prototype" subsection of Methods. The results of the calibration and the spatial resolution obtained with light intensity images, as well as a function of the number of counts, are shown in Fig. 2. The reconstructed 3D position of the light source was compared to its nominal position on the stage. Justified by its size, significantly smaller than the nominal lateral resolution of the PLATON-prototype, the source was assumed to be point-like. The spatial residual of the depth, 2D and 3D position of the light source is defined as the difference between the true ("$t$") and the reconstructed position ("$r$"):

$$\Delta_{depth} = \sqrt{\left(z_r - z_t\right)^2} \tag{1}$$

$$\Delta_{rad} = \sqrt{\left(x_r - x_t\right)^2 + \left(y_r - y_t\right)^2} \tag{2}$$

$$\Delta_{3D} = \sqrt{\left(x_r - x_t\right)^2 + \left(y_r - y_t\right)^2 + \left(z_r - z_t\right)^2} \tag{3}$$

where $z$ is the depth along the optical axis of the PLATON-prototype, while $x$ and $y$ are the lateral dimensions. The resolution is defined as the 68% 1-sided integral of the residual distribution. The depth, 2D lateral and 3D resolutions in the regime of fully-illuminated frames, i.e., when the light field is captured to the best capability of the camera, are, respectively, 4.2 mm, 1.8 mm and 4.5 mm for the validation sample and 2.2 mm, 0.9 mm and 2.4 mm for the calibration sample. The difference in resolution between the validation and calibration data was found to be affected by remounting the camera on the movement stage between the calibration and validation data runs. This can be seen in the shift in

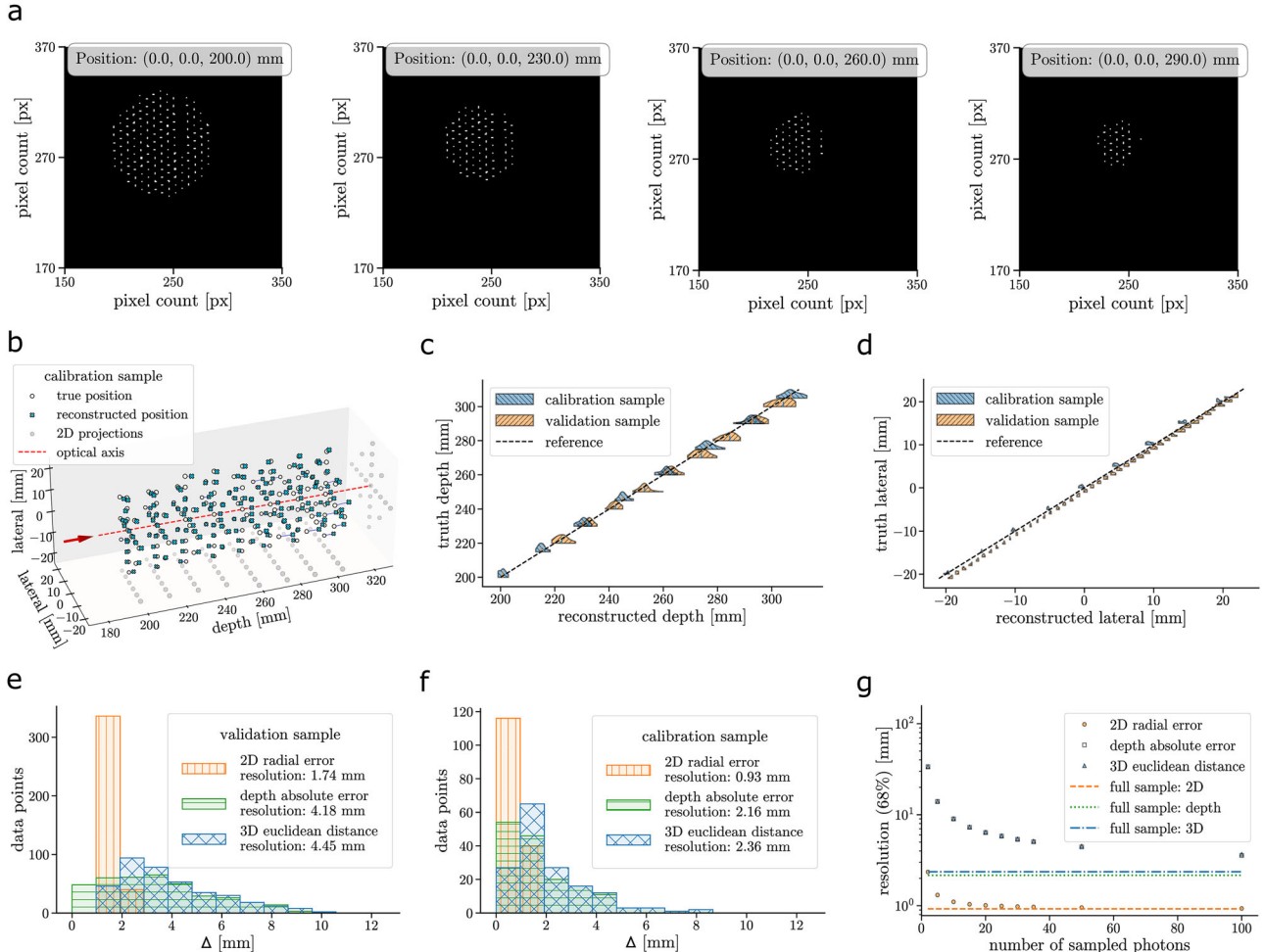

**Fig. 2 | Position reconstruction results for a point-like light source with the PLATON-prototype. a** Cropped images of the data taken during the calibration measurements. Visible is the calibration point source at different depths, on the optical axis; in classical imaging, the number of micro-images is used to infer the depth of the imaged objects. **b** True (white circles) and reconstructed (blue crosses) positions of the back-lit point light source with respect to the PLATON-prototype. Grey circles indicate the steps done by the moving stage projected onto the two perpendicular planes. The "violin" plots of the reconstructed depth (**c**) and lateral position (**d**) of the light source are shown. The dashed line corresponds to the lateral reconstruction in Fig. 2. Thus, we will refer to the calibration

the case of a perfect reconstruction. The probability density function of the source depth or lateral position is shown for the calibration (blue) and validation (yellow) samples. The one-sided distributions of the residuals of the measured 3D positions obtained from the validation (**e**) and calibration (**f**) samples are shown. **g** The depth (green), 2D radial (orange), and 3D (blue) spatial resolutions as a function of the number of photons sampled from the calibration sample are shown on bottom right. The dashed horizontal lines correspond to the resolutions for high yield obtained from (**f**).

the lateral reconstruction in Fig. 2. Thus, we will refer to the calibration sample to highlight the intrinsic resolution of the PLATON-prototype.

The spatial resolution was also studied as a function of the number of detected photons. A data-driven procedure was adopted: 1000 photon-starved images were obtained by sampling the pixel position of a certain predetermined number of counts, using the intensity image of the point light source at a specific position as a PDF. With 10 detected photons, the 2D lateral resolution falls below 1.2 mm. As expected, the 3D resolution is dominated by the depth one and becomes worse than 10 mm for fewer than 10 photons. The obtained spatial resolution is consistent with what is expected from the design specifications of the PLATON-prototype, including the reduced FoV of the camera, as well as with results obtained with the optical simulation (see Supplementary Methods Section 1). Using the simulated equivalent of the calibration sample, the extracted depth resolution is 2.25 mm, with a 2D radial resolution of 0.5 mm. When accounting for possible systematic uncertainties inherent in the stage and camera setup, these values are considered consistent with those obtained from measurements.

An alternative reconstruction method for point-like sources was also tested using a similarly acquired data sample. This method is based on evaluating the likelihood of the measured light patterns with respect to a reference data sample. A depth resolution of 1.5 mm was achieved along with a lateral resolution of 0.1 mm. With the sampled photon-starved images, the depth and lateral resolutions decreased to, respectively, below 10 mm and 0.3 mm for 5 photons. For fewer than 10 photons, depth ambiguities arise between points located along a line extending from the principal point, as these produce nearly concentric image patterns. For further details, see Supplementary Methods Section 2. It is worth noting that the likelihood study should not be interpreted as the achieved spatial resolution, as it would be impractical to build a PDF for every single possible point down to the relevant precision in a relatively large volume. On the other hand, the largely improved resolution compared to the reference method described above (better than a factor of 2 or 3 for more than 30 counts) indicates that the 2D frames provided by the PLATON-prototype contain more spatial information than what can be exploited with the developed post-processing method. Such a feature highlights that a more

advanced post-processing, such as one that uses a more detailed optical model than the approach described in the "Image post-processing method" subsection of "Methods," or deep-learning-based reconstruction as described in the "Neural-network based reconstruction" subsection of Methods, would further improve the 3D spatial resolution.

## Detection of ⁹⁰Sr electron events

The final measurement with the PLATON-prototype aimed to detect, with a plenoptic camera, single elementary particles in plastic scintillator on an event-by-event basis. A 50 mm × 50 mm × 50 mm block of EJ-262 plastic scintillator[67] (parameters are listed in Supplementary Methods Section 1) was exposed to a ⁹⁰Sr source, placed on the top side. The electrons exiting the container of the source can carry an energy up to a maximum of 1.5 MeV. In order to maximise the photon collection, we had to position the scintillator block about 5 cm from the main lens of the PLATON-prototype, just outside the optimal depth of field (150 mm), and hence out of focus. This is limiting the angular resolution on the photon direction. To certify the reconstruction of the electron position, two different datasets were collected, with and without the ⁹⁰Sr on top of the scintillator block. We name the two datasets, respectively, as ⁹⁰Sr and background. To decrease the probability of dark counts, the PLATON-prototype was placed in a thermal chamber at a temperature of −5 ℃, reducing the median dark count rate (DCR) to 0.002 cps/μm² (0.4 cps/pixel). The background sample allows us to identify noisy pixels, i.e., with a DCR higher than 0.5 cps/pixel, which were masked at the post-processing stage, further reducing the SwissSPAD2 photodetection efficiency (PDE) down to 2.5%, however, increasing the signal-to-noise ratio. Thus, given the limited light yield of the prototype (f/2.4, active area reduced by the glue), only frames with at least four counts were selected, as these are more likely to contain at least two scintillation photons, necessary to fit a point in the object space. In the post-processing described in the Methods section, candidate photons were identified if the relative minimum distance was less than 3 mm. Out of 9 frames with four counts found in the ⁹⁰Sr sample, only four events were selected upon the convergence of the post-processing. As shown in Fig. 3d, for each event, the reconstructed position of ⁹⁰Sr was closer than 20 mm to its true position, dominated by the depth resolution, as discussed above.

This result is consistent, within statistical uncertainty, with the spatial resolution obtained for a low number of photons when accounting for the distance between the main lens and the scintillator, which is not in focus, as shown in Fig. 2.

The same analysis was done on both the ⁹⁰Sr and the background sample with frames containing 3 counts. The rejection rates were, respectively, 61% and 81%. In the latter two cases, the distribution of the distance from the ⁹⁰Sr nominal position is evenly distributed across the entire depth range. Event displays of the four ⁹⁰Sr candidates can be found in Supplementary Fig. 1. From a toy Monte Carlo simulation of the background sample, we expect 0.17 background events with 4 or more counts to be reconstructed between 0 and 20 mm from the true ⁹⁰Sr position.

## Simulated particle detection with the PLATON-10cm detector

As a physics study case, we chose the detection of GeV muon neutrinos. In fact, the PLATON concept could be suitable for the near detector of future accelerator long-baseline neutrino oscillation (LBL) experiments, which will search for leptonic charge-parity violation and determine the neutrino mass ordering[1-3]. In these experiments, the precise measurement of the neutrino-nucleus cross section is crucial for the reduction of the key systematic uncertainties, and it relies on the high-resolution tracking and calorimetry of final-state low-momentum hadrons (e.g., multiple protons down to 200 MeV/c). We studied the performance of a small PLATON module

consisting of two optical arrays, each of four plenoptic cameras, pointing to two orthogonal faces of a 10 × 10 × 10 cm³ block of EJ-262 plastic scintillator (parameters are listed in Supplementary Methods Section 1), which we call PLATON-10 cm. Performing the analysis described below would be challenging for a larger detector volume with the computing power currently available. Since the attenuation length is way longer than the scintillator volume, a perfect transparency was assumed. This assumption is justified even for larger volumes due to the availability of organic liquid scintillators with analogous light output but attenuation lengths of the order of 20 m or greater[63,68]. The simulated detector is shown in Fig. 4a. Owing to the results obtained with the PLATON-prototype, we designed the PLATON-10cm detector unit (a single camera). The adopted strategy consisted of optimising the lateral resolution, with a depth resolution of 1 mm over a depth of field of 10 cm. Exploiting the depth resolution to match the two orthogonal views, the result is a 3D spatial resolution that mainly belongs to the sub-millimetre lateral one, especially in relatively low-multiplicity events such as neutrino interactions. Both the MLA and the main lens were simulated with f/1. The diameter of the main lens is 50 mm, while that of the microlenses is 1 mm. Realistic parameters of an optimised SPAD array sensor, which we refer to as PLATON SPAD, were simulated. The PDP (40% peak at 500 nm), simulated as in Supplementary Methods section 1 and the pixel pitch of 25 μm with a 60% geometrical fill factor, amount to a PDE of 24%. We neglect the tiling between adjacent photosensors. With all efficiencies accounted for, 3670 photons were detected per event. The SPAD time structure shown in Fig. 4b was simulated. Each count is measured with a 200 ps resolution timestamp. R&D on PLATON SPAD is ongoing to achieve the parameters above[69].

## Single-point spatial resolution of an array of multiple cameras

Before studying the capability of a PLATON-10 cm module to detect neutrinos, the 3D spatial resolution for a point light source equivalent to 1 MeV energy deposition was evaluated. Of the 10,000 photons produced, on average, 141 photons were detected. The 3D spatial resolution was found to be 0.3 mm. For completeness, we compared plenoptic cameras with classical cameras, obtained by removing the MLA and adapting the lens-sensor distance, focusing the camera on the centre of the scintillator. Here, an average of 149 photons was detected per event. While a single classical camera cannot precisely reconstruct the 3D position of the light source, we found that the resolution provided by multiple classical cameras is about four times worse than that of plenoptic cameras. Comparing a setup with two plenoptic cameras to a setup with two classical cameras, we see a significant difference in reconstruction efficiency, indicated in Supplementary Fig. 2. While the two plenoptic cameras are able in all events to reconstruct the 3D origin of the point source, when the event only falls in the view of a single classical camera, the reconstruction fails, leading to around 23% of failed reconstruction attempts. On the other hand, we note that the size of the scintillator block is not that large and does not require a very long depth of field (DoF), making the performance of an array of classical cameras not too far from that of plenoptic cameras. As we will see later in this section, the resolution of a system of classical cameras deteriorates compared to plenoptic cameras for a m³ volume. Thus, we focus on the PLATON-10cm configuration with plenoptic cameras, as described in the previous section. For more details, we refer to Supplementary Methods section 2.

## Neutrino detection in PLATON-10cm

The neutrino flux of the T2K LBL experiment[70,71] was simulated parallel to both sensor planes, and uniformly in the scintillator volume with the NEUT 5.5.0 event generator[72]. Assuming that an external detector, such as a time projection chamber with a spatial resolution and angular

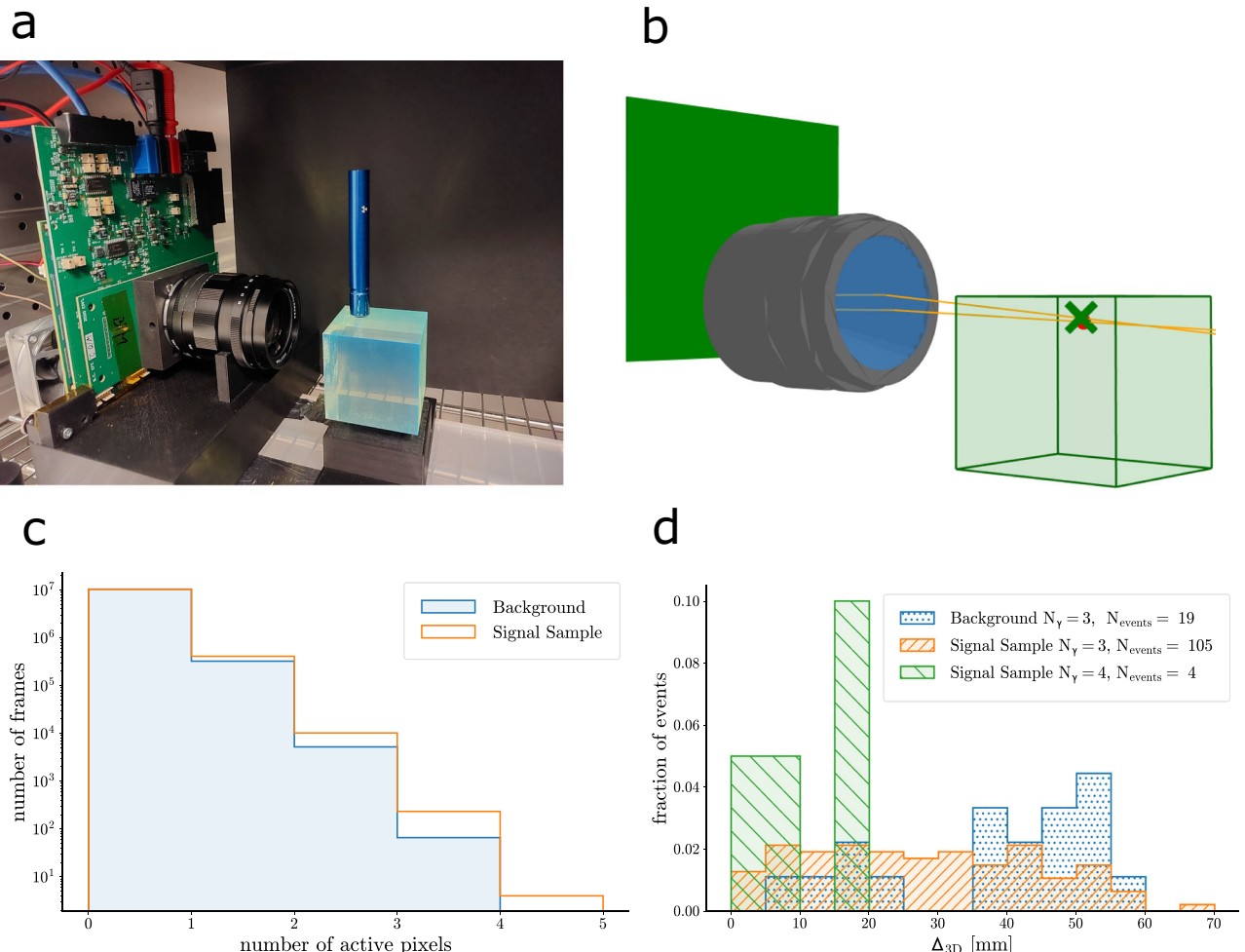

**Fig. 3 | Detection of $\beta$ electrons from a $^{90}$Sr using the PLATON-prototype.**
**a** PLATON-prototype pointing to the scintillator block exposed to the $^{90}$Sr positioned on the top face. **b** A candidate electron event in the scintillator (green volume) selected from the $^{90}$Sr sample. The prototype main lens, the reconstructed ray trace of the scintillation photons (orange lines), the nominal position of the source (red dot) and the reconstructed position of the electron (green cross) are

shown. Other $^{90}$Sr events can be found in Supplementary Fig. 1. **c** Distribution of the number of counts per frame for the $^{90}$Sr (orange) and the background (blue) samples. **d** Residual of the reconstructed $^{90}$Sr electron position from the $^{90}$Sr sample with 4 counts (green), 3 counts (orange) and from background sample (blue) with 3 counts.

resolution of 1 mm and 0.05 rad, respectively, in a magnetised volume, would be able to detect and identify the muon escaping the scintillating volume, we simulated a sample of charged-current (CC) neutrino interactions, disregarding the contribution of neutral-current (NC) events.

We developed a custom neural network based on the transformer architecture, commonly used in generative large language models, to efficiently capture the hyper-dimensional correlation between the detected scintillation photons. Details on the neural network are given in the "Neural-network based reconstruction" subsection of "Methods". We evaluated the particle tracking performance by computing the distance between each point (Geant4 output, see Supplementary Methods Section 1) along the truth trajectory of the particle and the closest reconstructed point (Fig. 4d, e). Overall, an average 3D-tracking resolution of approximately 190 µm is obtained, with a range of 150 µm to 340 µm. When considering the distance between each reconstructed point and the closest truth point, the resolution becomes more consistent, with a narrower range of 160 µm to 230 µm, though the average remains unchanged. The resolution is smaller than 200 µm for events with three particles or fewer. When particle ranges are less than 3 mm, the resolution averages 180 µm. Overall, the results demonstrate that a tracking

accuracy well below 1 mm is achievable within a sizeable PLATON module. This is in agreement with the results of Supplementary Methods Section 2 and the single-point spatial-resolution study of the PLATON-10cm module. A pattern recognition algorithm was developed to group the photon origins into different clusters that are then assigned to different particles. The results of the pattern recognition are shown in Fig. 5. The interaction vertex resolution for the full sample of CC interactions is 0.42 mm. The average difference between the number of identified and true particles is 0.25 with a standard deviation of 0.74. More details about the implementation of pattern recognition and particle identification can be found in Supplementary Methods Section 4.

Two samples of candidate neutrino CC interactions were selected, featuring either a muon and a proton with no pions in the final state (CC $1\mu0\pi1p$), or a muon and two protons with no pions (CC $1\mu0\pi2p$). The first one is the most common final-state topology below 1 GeV, thus the dominant channel at the T2K and Hyper-Kamiokande LBL experiments, and is mainly affected by CC quasi-elastic (CCQE) interactions, accounting for approximately 37% of all events. The second sample is a golden channel to characterise nuclear processes, such as 2 particle - 2 holes (2p2h) events or final state interactions (FSI) that can introduce biases in the reconstruction of the neutrino energy. The

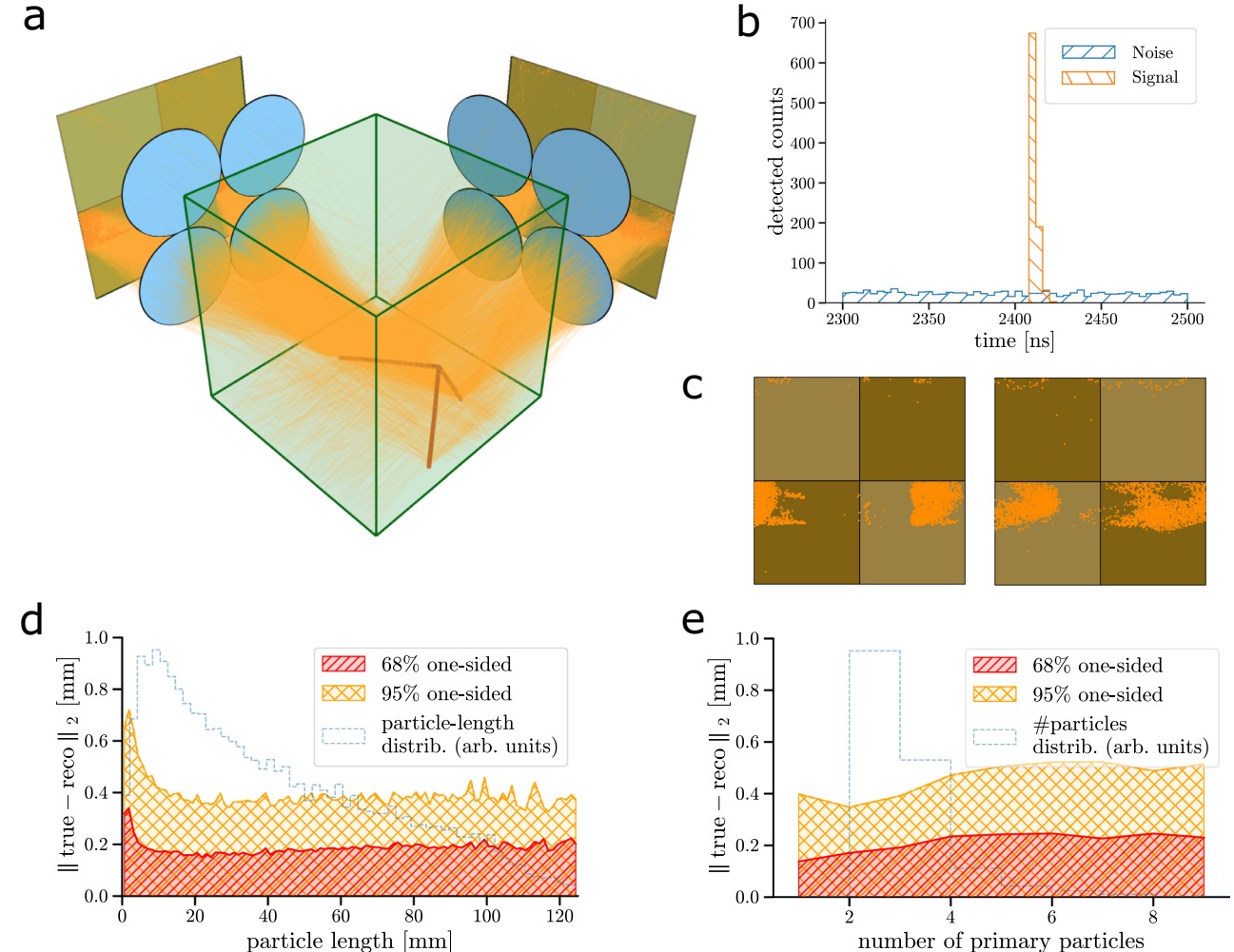

**Fig. 4 | Simulation and evaluation results of the proposed PLATON-10 cm detector module. a** Simulated interaction of a muon neutrino in the PLATON-10cm module. Both the truth particle tracks and the ray traces of the detected photons along with their arrival positions at the SPAD array are shown in different tones of orange. The count time distributions for signal photons (orange) and dark counts (blue) are shown in (**b**), while **c** illustrates the frames collected by the SPAD array sensors. 3D spatial resolution as a function of the particle range (**d**) and the number of particles in the event (**e**). Respectively, the distribution of the particle range (**d**) and of the number of particles per event (**e**) is shown (dashed light blue).

proper theoretical modelling of these processes is a source of significant uncertainties in LBL experiments. Thus, the precise detection of CC $1\mu0\pi2p$ events is crucial for the precise measurement of neutrino oscillations. More information can be found in Supplementary Methods Section 4, together with the details of the neutrino event selection.

To demonstrate the capability of a PLATON-10cm detector module to detect neutrinos, we focus on the selection of both CC $1\mu0\pi1p$ and CC $1\mu0\pi2p$ events, noting that the latter are notoriously more challenging to identify due to the presence of higher multiplicity. From Fig. 5f, one can see that the momentum of stopping protons reconstructed from the range is inferred with a resolution better than 10% across almost the entire range, approximately 5% at the energy flux peak, and close to 10% only below 200 MeV/c. The CC $1\mu0\pi2p$ selection purity is 90% up to 1.4 GeV/c momentum of the most energetic (leading) proton. The small background contamination comes mostly from events with higher proton multiplicity, typically produced with lower momenta. The total selection efficiency is 78%. It exceeds 80% for leading proton momenta above 300 MeV/$c$, but drops below 50% for momenta above 1.1 GeV/$c$, where the muon and the proton become more collinear. The proton reconstruction efficiency, shown for the CC $1\mu0\pi2p$ sample, exhibits a 50% threshold at approximately 215 MeV/c.

For the case of CC $1\mu0\pi1p$ events, both the selection efficiency and the purity reach 92%. The direction of single protons is determined with an angular resolution of 1.5°.

The same analysis has also been conducted for a system of classical cameras, as reported in Supplementary Methods Section 4. When compared to plenoptic cameras, similar conclusions can be drawn, as discussed in the single-point spatial-resolution study of the PLATON-10cm module and in Supplementary Methods Section 2.

**Single-point spatial resolution at 1 m³ scale**
Finally, the spatial resolution for a point light source in a PLATON detector with a $1 \times 1 \times 1$ m³ organic scintillator has been studied, named PLATON-1m. Due to limited computational resources, we were unable to perform a neutrino simulation study similar to that of the PLATON-10cm module, but we leave it for future work. The same analysis as in the single-point spatial-resolution study of the PLATON-10cm module was performed. As shown in Fig. 6, a 3D spatial resolution of 3.7 mm can be achieved for an energy loss of ~1 MeV (10,000 photons generated). For reference, this resolution corresponds to an effective segmentation of approximately 7.5 mm. Unlike a segmented detector, it improves up to 1.5 mm for an energy loss of ~10 MeV (100,000 photons generated), with an effective segmentation of 3 mm.

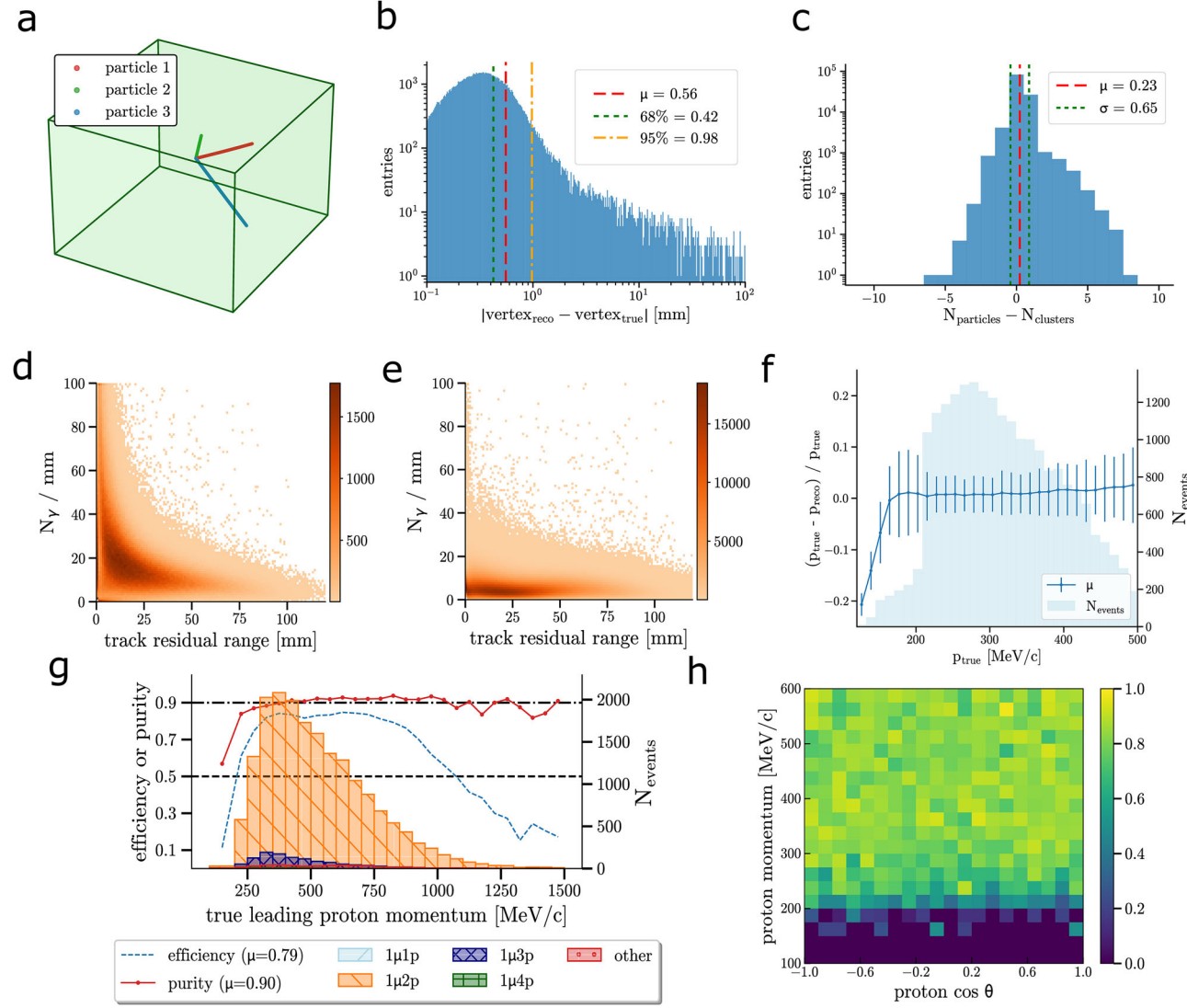

**Fig. 5 | Simulated muon neutrino event analysis results. a** Particle tracks reconstructed from a CC $1\mu0\pi2p$ neutrino interaction (a muon and two protons with no pions) after applying the image post-processing and the pattern recognition. **b** Residual of the reconstructed neutrino vertex position for proton events. **c** Difference between the number of particles produced by the neutrino interaction and the number of reconstructed clusters for muon events. The measured energy loss by a proton **d** and a muon **e** along the corresponding reconstructed track for all the events. **f** Proton momentum resolution (mean and 68% std) as a function of the stopping proton true momentum for selected CC $1\mu0\pi2p$ events (dark blue line). The true momentum distribution (light blue) of the protons from all the neutrino events is also shown. **g** True momentum distribution of the leading proton in the CC $1\mu0\pi2p$ selected sample, including efficiency and purity. **h** Proton reconstruction efficiency (colour map) as a function of the stopping proton true momentum and angle after applying the CC $1\mu0\pi2p$ selection.

The same analysis was performed with arrays of classical cameras. As anticipated above, this time we found that for a $1\,m^3$ size detector, the plenoptic system has a spatial resolution almost four times better than that of classical cameras.

With a few improvements, a sub-millimetre spatial resolution in a $1\,m^3$ scintillator could be achieved. First, the optical parameters have not been optimised for such a volume but have been kept the same as for the $10 \times 10 \times 10\,cm^3$ module. Moreover, the used post-processing method adopts a simplified parametrisation of the MLA. On the other hand, we also observed that the neural network described in the "Neural-network based reconstruction" subsection of "Methods" allowed for an additional improvement in spatial resolution. Thus, one might expect a similar improvement also in the tonne-scale PLATON-1 m. It is interesting to note that a smaller pixel pitch, though technically challenging, would allow for better constraint of the light field. Preliminary studies showed 30%

improvement in spatial resolution for a $1\,\mu m$ pitch without any adaptation of the optics.

## Discussion

In this work, we propose a change of paradigm in the high-resolution detection of elementary particles in dense scintillator-based detectors. The key is the use of imaging photosensors with single-photon subnanosecond time resolution, such as SPAD array sensors, integrated into a system of plenoptic cameras to provide unambiguous 3D images of elementary particles interacting within a monolithic volume of scintillator. We referred to this detector concept as PLATON.

We built and successfully characterised the spatial resolution of the PLATON detector prototype, a plenoptic camera instrumented with a SPAD array sensor, and we developed a post-processing method suitable for photon-starved images. We also succeeded in reconstructing the position of $^{90}Sr$ electrons on an event-by-event basis. All

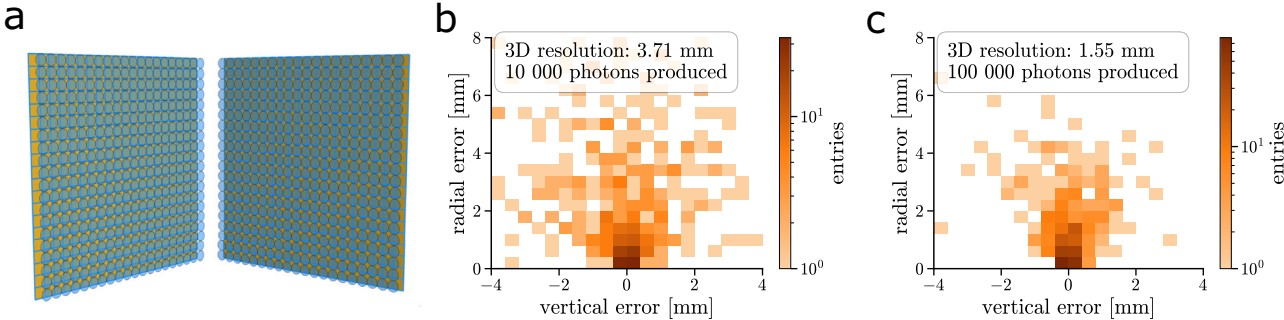

**Fig. 6 | Setup and reconstruction of a tonne-scale version using the PLATON detector concept. a** The geometry of a tonne-scale $1 \times 1 \times 1\mathrm{m}^3$ PLATON-1m detector. The plenoptic cameras are shown in blue. The same optics configuration as used for the PLATON-10cm point-source study is adopted, resulting in a total of 800 plenoptic cameras, each with about $4 \times 10^6$ pixels (~$3.2 \times 10^9$ pixels in total). The spatial resolution in cylindrical coordinates for a point light source emitting **b** 10,000 photons and **c** 100,000 photons is shown.

the results are in agreement with our simulations, which gives us confidence in designing a realistic PLATON detector for neutrino detection.

Building on the experience gained through prototyping, we carried out a simulation-based analysis of muon neutrino interactions in a PLATON detector, employing an improved yet realistic design. We developed a high-performance image post-processing based on deep learning and complemented by pattern recognition algorithms. A tracking resolution down to 200 μm, along with high purities and efficiencies in the selection of neutrino interactions with multi-proton final states, was achieved.

The extrapolation to a 1-tonne scintillator detector showed a spatial resolution of a few millimetres for point light sources. Although it is already competitive with state-of-the-art plastic scintillator detectors, we expect more sophisticated image post-processing, such as deep learning, or a plenoptic system optimised for a metre-scale volume, to be promising solutions for achieving sub-millimetre spatial resolution. Future developments, such as optics optimised for metre-scale volume and smaller pixels, will further improve the spatial resolution. This will be the topic of future dedicated work.

We compared the simulation results with those obtained using a system of classical cameras and found that plenoptic cameras outperform classical systems, regardless of the scintillator volume size.

The PLATON detector configuration offers broader applicability beyond accelerator-based neutrino studies, which were adopted here as a case study. Its calorimetric capability, combined with high spatial resolution, could make this concept potentially suitable for scintillator-based neutrino-less double-beta decay ($\nu$-less $\beta\beta$) experiments and make it worth studying. Examples could be ref. 73 or large-scale detectors such as refs. 74–77 (see also ref. 78 for a comprehensive review). Furthermore, following the design philosophy outlined in ref. 48, the PLATON concept appears promising for application in very-large-scale neutrino detectors[68,79–81]. Similarly, it may enhance the reconstruction of light rings characteristic of Cherenkov radiation[82,83]. Compared to liquid argon time projection chambers (LArTPCs), currently the only technology capable of 3D tracking particles produced by neutrino interactions with O(mm) resolution in giant volumes, the PLATON technology would profit of its much faster response. Thus, it is ideal for the detection and the reconstruction of the time-of-flight of fast neutrons produced by neutrinos. All would be achieved without the need of big cryogenic infrastructures. On the other hand, its scalability to the kilotonne scale achieved by LArTPCs will have to be proven.

To conclude, our work paves the way towards high-resolution particle tracking and calorimetry in unsegmented volumes. The development of SPAD array imaging sensors, or other photosensor technologies with analogous features, is the key. We are currently conducting R&D to develop a SPAD array sensor that meets the requirements discussed in this article, specifically a PDE similar to that of high-dynamic range SiPMs[32] and single-photon sub-nanosecond time resolution.

Owing to high-spatial resolution in large and dense active volumes combined with an excellent time resolution, the PLATON concept therefore naturally opens up new areas of application for deploying plenoptic cameras for imaging, including positron-emission tomography (PET), neutron radiography, muon tomography, synthetic computed tomography (CT), and proton CT.

## Methods

### Image post-processing method

The detection of elementary particles in scintillator requires single-photon detection. Thus, photon-starved images replace the more traditional ones based on light intensity. A post-processing method was developed to backtrace all the photon rays from the image plane (sensor pixels) to the object space (light source).

Post-processing algorithms for plenoptic systems are based on parallax detection and require pattern matching between images observed by different micro-lenses[84]. In this reconstruction regime, a single point source would create a sub-image in a set of neighbouring micro-lenses. If the point source sits in the focus region of the plenoptic camera, each microlens images a single pixel, while a point source out of focus would create a blurred image in each participating microlens. The number of active microlenses is known as Virtual Depth (VD) and is proportional to the physical distance of the object from the main lens, allowing for depth estimation. However, we decided not to adopt this technique as it cannot be easily applied to photon-starved frames, as it would lead to the pattern matching failing. Thus, we require an optical model (see Supplementary Methods Section 1) to project each detected photon through the optics into the object space. Post-processing, based on that of a pinhole camera, is also used for plenoptic cameras in light-intensity imaging[85]. In this work, we adopt the same principle to the case of photon-starved images and, thus, developed the following method: the ray trace of every photon that was detected by a pixel of the sensor (a count) is forced to pass through the centre of the closest micro-lens, and interpolated in the object space. The optical model described in Supplementary Methods Section 1 was used for the propagation of the ray tracing: the MLA was parametrised as an array of pinholes, and each photon was assumed to have impinged at the centre of the pixel. Outlier ray traces, mainly generated by noise, but also by the adopted approximation, were iteratively rejected until either a predefined minimum number of rays with the relative closest distance was obtained or the distance between each

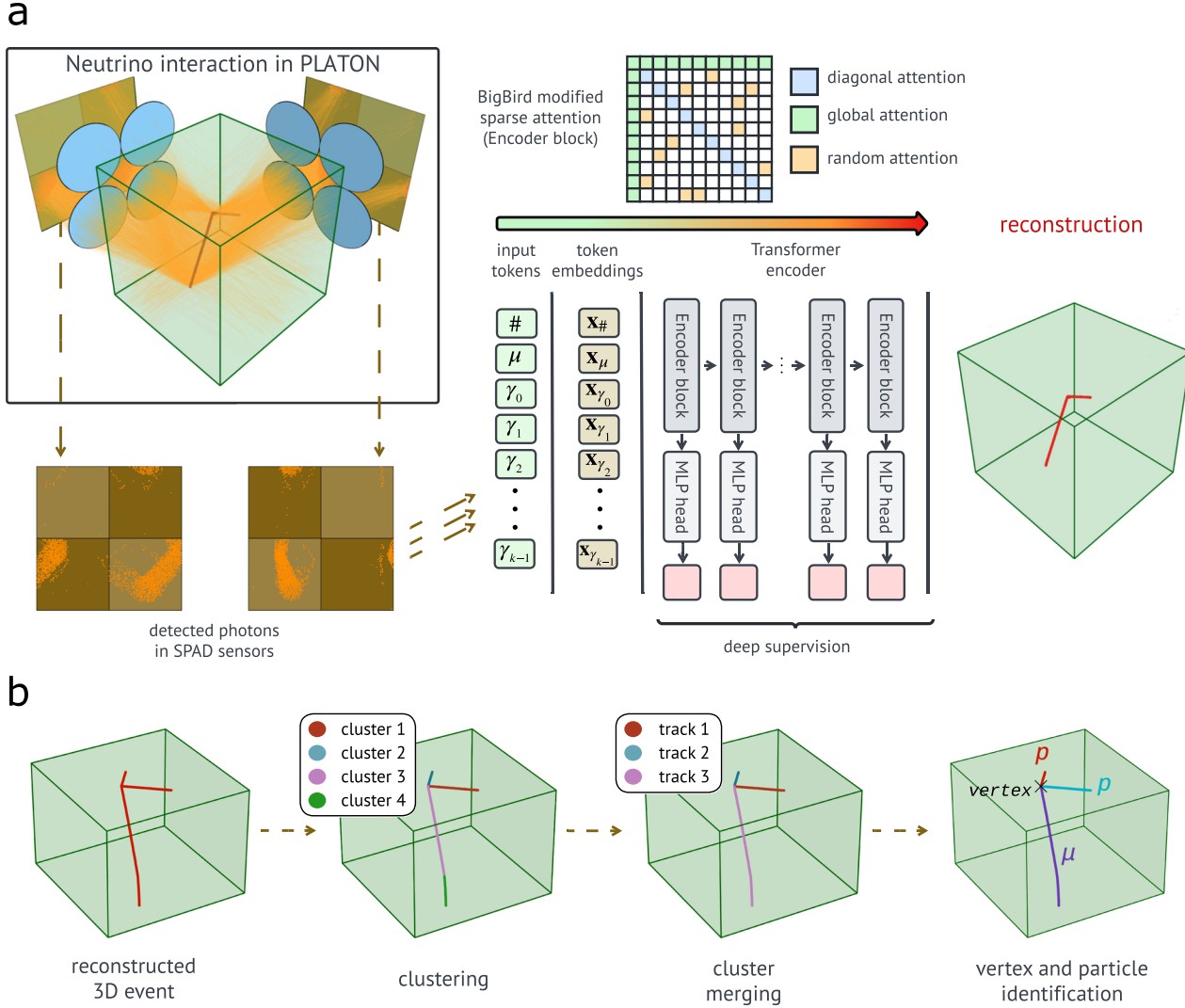

**Fig. 7 | Neural event reconstruction and pattern-recognition pipeline. a** Neural-network-based reconstruction workflow. From left to right: the detected 2D photon coordinates are grouped into a sequence of tokens $\gamma$, with two additional tokens appended: #, indicating the total number of photons, and $\mu$, optionally encoding the muon's exiting direction and position. Then, a micro-lens embedding identifying the optical element that detected the photon is summed to the photon coordinate embeddings, and a token type embedding (indicating whether the token represents a photon coordinate, photon count, or muon exiting information) is summed to each token embedding. A modified BigBird Transformer encoder[88] processes the resulting combined embeddings, which leverages diagonal, global, and random sparse attention mechanisms. Finally, multi-layer perceptron (MLP) heads with deep supervision are used to perform the event reconstruction. **b** Pattern recognition steps. From left to right: reconstructed hits are grouped into clusters; collinear and nearby clusters are merged together; finally, particle identification and vertex reconstruction are performed.

leftover ray was below a certain threshold. The interpolation with the remaining ray traces was used to estimate the position of the light source.

The constraint on the direction of the photon rays induced by the pinhole approximation makes this method practical, as the computational complexity scales linearly with the number of photons detected. Eventually, the triangulation is performed among the propagated rays in the object space, based on the minimal distance between different rays.

**Calibration of the PLATON-prototype**

The post-processing of the 2D images relies on the accurate calibration of the optical model of the prototype, that is, the data-driven inference of the model parameters discussed in Supplementary Methods Section 1. First, the lateral position of each micro-lens is obtained by illuminating the PLATON-prototype without the main lens with a source of parallel light rays. This was achieved using a low-power laser mounted on a custom-made motorised stage, which could be translated parallel to the MLA plane. A single greyscale intensity image was obtained from 4096 frames and used to determine the centre of each micro-lens. The mean distance between the centres of two adjacent micro-lenses was computed to derive the parameter $D_{\mu}$. The position of the micro-lenses could not be determined for the entire area of the MLA, as the glue partially covers that region (see Fig. 1). Consequently, that part of the image was not used in the post-processing. This issue reduced the field of view of the PLATON-prototype. Imaging a parallel-light source is not sufficient to infer the remaining parameters of the optical model. Hence, we acquired a dataset consisting of 12-bit images of a back light-illuminated 50 µm pinhole at 168 different positions within an imaging volume of $4 \times 4 \times 12 \text{cm}^3$. Each image was obtained by stacking 4096 frames, each recorded with a 10 µs gate. The motorised stage, by design, allowed the light source to move in steps of 10 µm. Measurements where the light source fell outside

the camera's field of view (FoV), reduced by the glue, were removed from the dataset. In total, 160 images were used to fit the parameters of the optical model using a custom genetic algorithm, in which the post-processing method was iteratively applied, and the distance between the reconstructed and true positions was minimised.

The nominal values of $B$ and $f_L$ provided by the manufacturers were used as seeds in the fit, while the parameters of the MLA, previously measured with the parallel-light source, were kept fixed.

The optical simulation (see Supplementary Methods Section 1) was used to reproduce images from a 100 μm point source at the same 168 positions as those of the acquired dataset. The focal length of the micro-lens array, $f_{MLA}$, was fixed to its nominal value in the datasheet of the MLA. Then, the simulated images were compared to the corresponding data samples, and good agreement was observed, indicating that the model provides a reasonable representation of the camera's optical stack.

In addition to the calibration sample, an independent dataset—referred to as the "validation" set—was acquired and used to evaluate the performance of the post-processing method. In this case, 387 different positions were taken with a depth spacing of 10 mm over a range of 80 mm. The lateral spacing was 1 mm, ranging from −21 mm to 21 mm, along one of the lateral axes. The results of the calibration, as well as the obtained spatial resolution on light intensity images as a function of the number of detected photons, are shown in Fig. 2. The validation sample exhibits a spatial resolution slightly inferior to that of the calibration sample. We realised that this was due to a misalignment caused by the disassembly of the experimental setup between the acquisition of the two data samples.

### Neural-network based reconstruction

The dataset used for the neural network studies comprised 1M CC-inclusive muon neutrino interactions generated as described in the neutrino-detection study of the PLATON-10cm module. The simulation of the detector setup is described in the "Simulated particle detection with the PLATON-10 cm detector" subsection of Results. Reconstructing 3D neutrino interactions within this setup is inherently challenging. SPAD arrays are arranged orthogonally on two sides of the scintillator block. Each interaction produces a photon-starved pattern, typically comprising between a few tens and several thousand detections across the SPADs. The task is to infer the spatial origin and topology of the interaction using only these sparse observations.

To achieve this, we opt for a transformer-encoder-based architecture (see Fig. 7a) to fully exploit the spatial correlations among detected photons through the transformer's attention mechanism[86,87]. Unlike traditional architectures that rely on local connectivity or fixed sequence order, the transformer's attention mechanism enables dynamic interactions between all elements in the input. This is particularly beneficial for our application, where the input consists of sparse and spatially-distributed photon detections and where meaningful correlations may exist between distant regions of the detector. In our setup, each photon is treated as a token, embedding its 2D position on the SPAD array, the identifier of the micro-lens it passed through, and the corresponding SPAD sensor ID. Additionally, the input sequence includes a dedicated token encoding the number of sampled photons in the event and one token for each muon exiting the scintillator block (typically one), which contains the 3D exit point and direction vector. The maximum sequence length is fixed at 1,024 tokens. During training, we randomly select $\lfloor 0.9N \rfloor$ of photons per event (with $N$ being the number of photons in the event, further capped by the maximum sequence length) and fill the remainder of the sequence with a padding token, which is ignored during training. This stochastic subsampling means that, from the model's perspective, the same physical event is seen with a different subset of photons at each epoch, and thus acts as an input-level regularisation/data-augmentation mechanism, reducing overfitting to specific photon patterns and improving generalisation to unseen events. Given that each token already includes spatial information, explicit positional encodings are unnecessary. Early experiments showed that the network struggled to reconstruct events when the number of detected photons was significantly lower than the maximum sequence length. We hypothesise that this limitation arises from the model's lack of awareness of the photon count in each event. To address this, we include the aforementioned dedicated token encoding the sequence size, which resolves this issue by enabling the model to condition the number of observed photons.

A detailed description of the architecture, including the comparison with alternative approaches, the input tokenisation, the sparse attention mechanism, the loss functions, and the training procedure, can be found in Supplementary Methods Section 3. The pattern recognition and particle identification steps applied to the neural network output (Fig. 7b) are described in Supplementary Methods Section 4. An overview of this workflow is shown in Fig. 7a.

## Data availability

The minimum dataset necessary to interpret, verify and extend the findings of this study has been deposited in Zenodo: https://doi.org/10.5281/zenodo.18701772. This record contains a representative subsample of the dataset, with accompanying metadata/documentation, sufficient to reproduce the analyses reported in the paper.

## Code availability

The code used in this work is not publicly available at this time due to ongoing intellectual property protection efforts. It may be made available upon request, subject to appropriate terms and, where applicable, the execution of a non-disclosure agreement.

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

## Acknowledgements

This work was supported by the Swiss National Science Foundation under grantPCEFP2_203261. This research was also partially supported by the Swiss National Science Foundation (grant 20QT21_187716 Qu3D "Quantum 3D Imaging at high speed and high resolution"). Neural network training in this work used resources of the National Energy Research Scientific Computing Center (NERSC), a U.S. Department of Energy User Facility, under the AI4Sci@NERSC award DDR-ERCAP0034642. Additional training support was provided through the Swiss AI Initiative via a grant from the Swiss National Supercomputing Centre (CSCS), project ID a149, on the Alps system. We would like to thank Prof. André Rubbia from ETH Zurich for useful inputs and discussions and for providing access to laboratory equipment and facilities; Prof. Vincenzo Berardi at Politecnico di Bari, Dr. Umut Kose and Johannes Wüthrich from ETH Zurich for useful discussions; Arne Erdmann from Raytrix GmbH for helping to understand the functioning of the plenoptic camera prototype assembled at Raytrix and the use of the RxLive software; Prof. Wallny at ETH Zurich for providing access to his group's thermal chamber.

## Author contributions

D.S. conceived the PLATON detector, is the PI of the project funded by the Swiss National Science Foundation, and supervised every aspect of the project. T.D. was the main analyser and developer. The plenoptic system of the prototype was designed and built by Raytrix GmbH. E.C., C.B., and K.K. provided the SPAD array photosensor, assisted with its use, offered guidance, and supervised the project to ensure understanding of the results. T.D., T.W., and M.F. set up the tests in the laboratory. T.D. ran the experiments, developed the software and analysed the data. T.D., S.A.-M., and D.S. conceived the standard image post-processing method, and T.D. developed and tested it. T.D. developed and tested the software for the simulation of the detector. T.D. studied the detector configuration of the simulated physics experiments. Also, C.A. worked on the detector simulation. S.A.-M. conceived, developed, trained, and validated the neural-network-based image post-processing. N.B., S.A.-M., and T.D. performed the pattern recognition and data analysis of the simulated neutrino experiment. All the authors contributed to the writing of the paper. This document was prepared by Swiss Federal Institute of Technology-Zurich (ETH Zurich), in part as a result of the use of facilities of the U.S. Department of Energy (DOE), which are managed by The Regents of the University of California, acting under Contract No. DE-AC02-05CH11231. Neither The Regents of the University of California, DOE, the U.S. Government, nor any person acting on their behalf: (a) make any warranty or representation, express or implied, with respect to the information contained in this document; or (b) assume any liabilities with respect to the use of, or damages resulting from the use of any information contained in the document.

## Competing interests

The authors declare the following competing interests: T.D., S.A.-M., and D.S. are named inventors on a patent application filed by ETH Zurich related to the technology described in this article (status: pending). E.C. is a co-founder of NovoViz. NovoViz was not involved in this work or in the drafting of this paper. C.A., C.B., N.B., K.K., T.W., and M.F. declare no competing interests.
