## [Transparent Peer Review file · Nature Communications]

An ultrafast plenoptic-camera system for high-resolution 3D particle tracking in unsegmented scintillators

Corresponding Author: Professor Davide Sgalaberna

Version 0:

Reviewer comments:

Reviewer #1

(Remarks to the Author)

1. Summary and Main Contribution

The authors present a novel detector concept ("PLATON") that, for the first time, combines a plenoptic camera architecture with time-resolved SPAD sensing to enable three-dimensional reconstruction of individual scintillation photons inside an unsegmented scintillating volume. The manuscript reports:

A fully functional hardware prototype

Demonstration of sub-millimeter lateral resolution and millimeter-scale depth resolution

Event-by-event reconstruction of β -electrons from a ^{90}Sr source

Detailed Monte Carlo simulations extending to neutrino event topologies

Extrapolation to ton-scale detector concepts

Integration of transformer-based deep learning reconstruction

This represents a substantial technological advance over conventional optically segmented scintillator detectors and introduces a fundamentally new approach to optical particle tracking.

2. Novelty and Originality

The level of novelty is very high.

To my knowledge, this is the first implementation of:

A plenoptic optical system combined with single-photon time-of-flight resolution

Real-time photon-by-photon 3D reconstruction in a fully unsegmented scintillator

A unified framework merging:

Ultrafast optical timing

Computational plenoptic imaging

Likelihood-based ray reconstruction

Transformer-based deep learning

This positions the work clearly at the interface between high-energy particle detection, computational imaging, and ultrafast photonics. The conceptual advance is genuine and not incremental.

 From a novelty standpoint, the manuscript fully meets the standards of Nature Communications.

3. Technical Quality and Physical Soundness

Strengths:

(a) Real experimental prototype

The work is not purely simulation-based. The authors demonstrate:

A fully instrumented prototype

Calibration using 168 spatially distributed reference points

Direct measurement of:

2D and 3D spatial resolution

Photon statistics

Dark count rate effects

This establishes clear experimental credibility.

(b) Physically consistent simulation framework

Geant4 particle transport

Optical ray-tracing including lens and sensor response

Realistic SPAD timing parameters

Direct comparison between plenoptic and conventional camera approaches

(c) Real radioactive source measurements

The reconstruction of β -electrons from a ^{90}Sr source is a crucial proof of physical principle. Even with limited statistics, it demonstrates that:

The system functions with real ionizing radiation

Individual energy deposition tracks can be recovered

(d) Neutrino-scale physics case study

The application to GeV-scale neutrino interactions is convincingly presented:

CCQE and two-proton topologies

Momentum reconstruction at the $\sim 10\%$ level

Vertex resolution of ~ 0.4 mm

Limitations and Weaknesses:

Extremely limited experimental statistics

Only four fully reconstructed ^{90}Sr events

Sufficient for proof-of-principle, but statistically weak

Optimistic simulation assumptions

Ideal optical clarity

Neglect of:

Optical scattering

Surface imperfections

Lens aberrations in large volumes

Cross-talk effects in dense SPAD arrays

Deep learning model treated as a black box

No ablation studies

No comparison to classical tracking algorithms

Little discussion of generalization and overfitting

Limited interpretability of failure modes

Scalability remains purely conceptual

Multi-camera integration, synchronization, and mechanical tolerances are not demonstrated experimentally

Ton-scale detector performance rests entirely on simulation

 The physics and optical modeling are sound, but the technological readiness level remains in the advanced prototype + simulation regime.

4. Methodology and Data Analysis

The methodological framework is very strong and carefully executed:

Genetic-algorithm-based calibration

Likelihood reconstruction used in parallel to ray tracing

Systematic resolution studies versus:

Photon statistics

Event multiplicity

Direct performance comparison between plenoptic and non-plenoptic readout

This level of quantitative validation is fully consistent with high-impact journal expectations.

5. Clarity, Structure, and Presentation

Overall, the manuscript is well written and clearly structured, with only minor issues:

Some sections are extremely dense, particularly the deep learning methodology

Occasional repetition between Results and Discussion

A few long, technically overloaded sentences

All of these are editorial rather than conceptual problems and can be readily corrected.

6. Significance and Impact

This work is of high relevance across multiple research communities:

Neutrino and rare-event detector physics

Scintillator-based radiation detection

Single-photon sensor and time-resolved imaging technologies

The possibility of achieving sub-millimeter 3D tracking in cubic-meter-scale unsegmented detectors is potentially transformative. Applications extend to:

Neutrinoless double beta decay

Fast neutron imaging

Medical imaging (e.g., advanced PET)

 The interdisciplinary impact aligns extremely well with the scope of Nature Communications.

7. Suitability for Nature Communications

Yes — the manuscript is clearly suitable for Nature Communications.

It satisfies:

Strong conceptual novelty

Working hardware demonstration

Advanced simulation validated against experiment

Broad multidisciplinary relevance

However, it is not a Nature (flagship) paper, since:

No direct fundamental physics discovery is reported

No large-scale experimental detector is realized yet

The result is technological rather than a breakthrough in fundamental theory

 Nature Communications is exactly the correct target journal.

8. Key Critical Points Likely Raised by Reviewers

The most probable reviewer concerns will be:

Very limited radioactive-source data

Optimistic optical and detector simulations

Lack of deep learning robustness studies

Incomplete experimental demonstration of large-scale scalability

These are significant but do not undermine the validity of the core concept.

Justification for the Recommendation: Major Revision

While the manuscript presents a highly novel and technologically impressive detector concept with clear potential for broad impact, several key aspects prevent it from being accepted in its current form without substantial revision.

First, the experimental validation remains limited in statistical scope. The reconstruction of only four β -decay events from a ^{90}Sr source is sufficient to demonstrate proof-of-principle functionality, but it does not yet provide the level of robustness expected for a high-impact journal. A larger experimental dataset, or at least a more detailed statistical uncertainty analysis, is necessary to substantiate the claimed reconstruction performance.

Second, the simulation framework relies on optimistic assumptions, particularly regarding optical transparency, scattering, cross-talk, and large-volume optical imperfections. Since the scalability of the concept to meter-scale or ton-scale detectors is one of the core claims of the paper, the absence of these realistic effects weakens the quantitative credibility of the projected performance.

Third, the deep learning reconstruction pipeline is insufficiently characterized. The lack of ablation studies, benchmarking against classical reconstruction approaches, and analysis of generalization and failure modes makes it difficult to assess the

true reliability and robustness of the neural network-based results.

Finally, although the concept is highly promising, the engineering scalability to multi-camera, synchronized large detectors is only discussed at a conceptual level and not yet supported by experimental demonstrations of timing synchronization, mechanical tolerances, or long-term operational stability.

For these reasons, the manuscript requires a Major Revision to strengthen the experimental evidence, improve the realism of the simulations, and clarify the robustness of the reconstruction framework. Importantly, these issues are addressable with additional analysis and validation and do not undermine the originality or fundamental validity of the approach.

In summary, the work is well suited for Nature Communications, but it requires a Major Revision to reach the level of experimental and methodological maturity expected for publication.

(Remarks on code availability)

Reviewer #2

(Remarks to the Author)

The paper presents an interesting approach to measuring interactions in transparent scintillators using a camera concept based on SPADs and micro-lens arrays. The concept is verified by a small scale prototype. The results from this test are then fed into a simulation of a larger scale high-energy neutrino detectors, aiming to measure charged current interactions from neutrinos in the GeV range, similar to large baseline neutrino experiments like T2K and Hyperkamiokande. The optical setup and measurement principle is well described and relatively easy to follow (interruptions in the flow are presumably due to the magazine's editorial policy to have the methods section at the end). The methods employed follow an established pattern. The data are analysed using machine learning methods which by and large are well described. Several shortcomings are visible in the paper, mostly related to details and context. Not much information is given on scintillator properties, average light yield (apart from being "sparse"), light yields along the path, attenuation effects and homogeneity for large scintillator volumes, scaling effects for large volumes (admittedly a technical detail), photon propagation times, potential effects from reflection and scattering inside a less perfect scintillator etc. This lack of detail makes it hard to evaluate the presented use case. Also it is noted that other general detector parameters like the overall detection efficiency (as opposed to the presented reconstruction efficiency) or the achieved vertex resolution (presumably better for events with more tracks in the same origin) are missing. An outer tracking detector for the produced muon is assumed, but no parameters are given. It would be worth discussing how the parameters of this detector (which presumably has to match with a track inside the scintillator volume) influence the results and would be taken into account for the application in CC neutrino interaction studied in this paper. Again, it is assumed that the beam parameters for the simulation are taken from T2K, but more details would be helpful here, as would be a comparison of Monte Carlo truth with the reconstruction in terms of very fundamental tracking parameters (vertex, momentum, angular resolution). Other detector technologies are being developed addressing the same problem, notably large scale liquid Argon TPC and a more niche application using opaque scintillators. None of these are discussed in the paper, while a comparison especially with the former would certainly be worthwhile (if only to put the performance in context). Last not least, it is easy to conceive that this particular method will have interesting applications beyond high energy neutrino physics. The paper would benefit from an outlook into the wider applications of this ingenious technology.

(Remarks on code availability)

Reviewer #3

(Remarks to the Author)

Key Results

The paper presents a new concept for particle detection and reconstruction in bulk scintillating material using plenoptic cameras, also known as "light field cameras". The authors demonstrate the viability of this technology with data from a simple prototype, as well as simulated data of more realistic applications. This is an exciting venue for development and has the potential to become a very useful technology in particle physics that can outperform existing methods.

Validity

Overall, the presented results seem to be sound, but some broad claims of superiority of the new technology seem optimistic and the manuscript does not provide the necessary results to back such claims.

Other than that, there are mostly issues of clarity where some more information or a rephrasing of the explanations would be useful.

Significance

These are significant results. The paper presents the first investigations into a new method to build a high-resolution particle detector that has not only implications in particle physics, but also in related fields. While the authors prove the concept, a lot more work is needed to turn this into a performant detector. To our knowledge they do not contract findings in the literature.

Data, Methodology and Analytical Approach

No data or code was provided together with the paper. This is standard practice in our field as it is normally not possible for reviewers to redo-the analysis or verify the code due to its complexity. However, the methods described are appropriate for the analysis presented.

Clarity and Context

The detailed suggestions for improvement listed below aside, the paper is well-written and presents a meaningfully investigation into the performance of an interesting new technology.

Reference

The paper comes with an extensive list of references. We are not aware of missing references.

Our Expertise

Our expertise is in statistical methods, detector development, and data analysis in neutrino physics and related fields. We are able to adequately judge most of the content of the paper. However, we fear that we lack the necessary expertise to meaningfully judge the applied machine learning techniques or transformer architecture.

Detailed Improvements

ll 30-34) The authors point out the necessity of large numbers of read-out channels as a draw-back of existing detector technologies, implying that this new technology will be superior in that regard. But if the detector walls are covered in plenoptic cameras, with 10s of thousands of single addressable pixels each, doesn't that also mean an incredibly large number of "channels"?

ll 293-302) This paragraph is impossible to understand when reading the paper for the first time. What does "augmented with additional data points" mean? I think this paragraph should either explain the "likelihood" method in more details, or in fewer. Just saying, that the authors tested an alternative method based on the likelihood of the seen light patterns and then quoting the achieved resolutions should be enough here.

ll 339-342) The authors mention that the prototype had the scintillator cube "out of focus". But they fail to mention what that means for the reconstruction. Does it even matter, since the path of every single photon is reconstructed individually? Why was it not possible to re-focus the objective on the cube?

Fig 3) It looks like even with the "maximised light collection" the authors have only recorded a hand full of events with 4 photons. Is this consistent with the expected light yield when considering solid angle coverage and sensor efficiency? What is the total "exposure time" to collect these few events? I.e. how many beta rays were not detected, or are indistinguishable from the noise?

d) Is this the 3D residual? The authors went through the trouble of defining all sorts of distances in Eq 1-3. Why not use these defined names in the plots? Instead, everything is called "distance", "residual", or even more confusingly "true - reco" in different plots.

l 437) It is somewhat confusing that every single detector is called "PLATON", while there are three distinct setups discussed in the paper: the single-camera prototype, the eight-camera simulation, and the 1 m³ simulation. These should be given distinct names, so it is clear what is being talked about in each section.

Fig 4c) If the cameras are focussed on the cube, why are the collected images so smeared out? It's probably not the MLA, since the microlenses should be very small on the scale of the picture. Is it because the depth of focus is so narrow?

d & e) The y axis seems to be some squared distance between true and reconstructed position. Is it 2D or 3D? Looks like the square of some L² norm. But if it is squared, why is the unit "mm" and not "mm²"?

Fig 5g) Is this the momentum of the leading proton or of both? What about the 3rd and 4th protons in the "3p" and "4p" contamination? How can there be a "1p" contribution? How is the purity defined? Usually, a purity as function of true quantity makes no sense. Is this just the ratio of "2p" in this plot? The y axis label is a bit confusing, as it could be read as "ratio of efficiency to purity". Should the horizontal lines be part of the legend? Are those numbers somehow significant? Or would it be better to just have horizontal grid line in the plot.

Fig 6) I think it would be nice to mention how many cameras and how many pixels in total this setup has.

ll 654-656) Could the authors substantiate the claim that this technology is well-suited for neutrinoless double-beta decay? It seems to me like the most important aspect for that is the energy resolution, and it was not shown that PLATON is superior in that regard.

ll 658-664) Could the authors substantiate the claim that PLATON will be well suited for large scale detectors and improve the Cherenkov ring reconstruction? Covering a kiloton scale detector with plenoptic cameras seems like a quite expensive proposal.

ll 675-680) Could the authors substantiate their claim the PLATON has "enormous" potential in these applications? What makes it potentially better than existing solutions?

ll 821-823) By how much did the glue reduce the field of view? Did that matter at all for the presented results?

Eq 4) Why is the sum divided by the number of pixels? Just the sum alone gives the log likelihood of the given pattern. Why divide that by the number of pixels? It should not matter for the mere finding of a minimum though.

ll 885 & 927) Why is this called "fake data"? What makes this "fake" as opposed to any other simulations done in this work?

ll 1087-1079) What is this "regularisation" mentioned here? Why is it needed? What happens if it is omitted?

(Remarks on code availability)

Reviewer #4

(Remarks to the Author)

(Remarks on code availability)

Version 1:

Reviewer comments:

Reviewer #1

(Remarks to the Author)

Dear authors,

After reviewing the article again, I believe that it is suitable for publication in its current form and should be published immediately. I would like to thank the authors for their valuable contribution to the technology of scintillation detectors and look forward to further work. These are excellent results and I hope that the work will be continued soon.

Best regards.

(Remarks on code availability)

Reviewer #2

(Remarks to the Author)

Thank you very much for providing a second version of your text. I enjoyed reading the first draft very much and was pleased with the improvements made for the second version, which very much addresses some of the shortcoming identified in the original.

(Remarks on code availability)

The code has not been made publicly available, as stated in the paper. It would not be appropriate to request the code as part of the review process.

Reviewer #3

(Remarks to the Author)

We are largely happy with the revised manuscript. There are a couple of items, which we think have not properly been addressed:

1)

Fig 4d,4e,5b (maybe 8b, 8d), 8e, 8f are still not using the distances defined in eq. 1-3.

2)
In the answer to our comment regarding Fig 3, the authors answered that the number of events from the 90Sr source is consistent with the expectation, but they don't say how many events with more than 4 photons are expected from the source. (It is however clear that it cannot be explained with the background only.)

(Remarks on code availability)

Reviewer #4

(Remarks to the Author)

(Remarks on code availability)

REVIEWER COMMENTS

Reviewer #1 (Remarks to the Author):

8. Key Critical Points Likely Raised by Reviewers

While the manuscript presents a highly novel and technologically impressive detector concept with clear potential for broad impact, **several key aspects prevent it from being accepted in its current form without substantial revision.**

The most probable reviewer concerns will be:

Very limited radioactive-source data

Optimistic optical and detector simulations

Lack of deep learning robustness studies

Incomplete experimental demonstration of large-scale scalability

These are significant but do not undermine the validity of the core concept.

- We reply to these points above later with the same but more detailed comments.

First, the experimental validation remains limited in statistical scope. The reconstruction of only four β -decay events from a ^{90}Sr source is sufficient to demonstrate proof-of-principle functionality, but it does not yet provide the level of robustness expected for a high-impact journal. A larger experimental dataset, or at least a more detailed statistical uncertainty analysis, is necessary to substantiate the claimed reconstruction performance.

- We agree that the number of reconstructed \$^{90}\text{Sr}\$ candidates is limited. To quantify the statistical significance, we performed a toy Monte-Carlo study by randomly sampling frames from the background-only (dark-count) dataset and applying the same selection and reconstruction chain. Under these conditions, we expect 0.17 background events with \$\geq 4\$ counts that reconstruct within 20 mm of the nominal \$^{90}\text{Sr}\$ position. In addition, the prototype performance is supported by the calibration measurements reported in the manuscript (pinhole scans vs. detected-photon statistics), where both lateral and depth resolutions are quantified using the chief-ray and likelihood

approaches. We added a clarifying note in Sec. 2.2.1 (around L390) to make this connection explicit.

Second, the simulation framework relies on optimistic assumptions, particularly regarding optical transparency, scattering, cross-talk, and large-volume optical imperfections. Since the scalability of the concept to meter-scale or ton-scale detectors is one of the core claims of the paper, the absence of these realistic effects weakens the quantitative credibility of the projected performance.

We believe there is no issue with the scattering length because the size of the scintillator ($10 \times 10 \times 10$ cm³) is way smaller than the typical attenuation length in PVT-based plastic scintillator (about 400 cm). Moreover, this study shall be generalised to the case of liquid scintillator (as highlighted in the paper, L750) without any compromise to the detector design. In this case, the attenuation length (contributions of both absorption and scattering) is known to be well above 20 m, making such effect negligible for the simulated configuration.

We believe there should not be any particular issue related to crosstalk between individual pixels. For example, pixel-pixel crosstalk is also expected to be negligible (e.g. <0.2% in SwissSPAD2 chip used for the prototype).

Concerning optical imperfections, we agree these could be a valid concern for the actual implementation of plastic scintillator detectors. As an additional check, we performed post-submission measurements in plastic scintillator using laser excitation and did not observe a significant degradation of the measured spatial resolution (≤ 500 μ m for light-intensity line scans) for a $50 \times 50 \times 150$ mm³ volume. We cite this as supporting evidence, while noting that a full, dedicated study of systematics in large volumes is beyond the scope of the present manuscript and we plan to publish it separately.

Third, the deep learning reconstruction pipeline is insufficiently characterized. The lack of ablation studies, benchmarking against classical reconstruction approaches, and analysis of generalization and failure modes makes it difficult to assess the true reliability and robustness of the neural network-based results.

We agree that rigorous characterisation is essential to certify the validity of any neural network-based reconstruction.

Regarding benchmarking against classical approaches, we would like to clarify that in the higher-photon-density regimes and for the complex event topologies addressed here, standard classical reconstruction pipelines are not practical: the corresponding likelihood-based or iterative methods become computationally prohibitive and often suffer from convergence issues in such high-dimensional settings.

To nevertheless validate the reliability of the network, we benchmarked its baseline performance in controlled conditions against the point-like source results in Sec. 2.2 (Fig. 2). As now stated in the revised manuscript (Sec. 2.3.2, highlighted in red), the network reaches resolutions in these scenarios that are comparable to the optical limits

established in our earlier detector characterisation, indicating that it correctly learns the underlying optical response.

We further probe robustness and generalisation by quantifying the 3D resolution as a function of (i) particle track length (Fig. 4d) and (ii) event multiplicity (Fig. 4e). These studies show stable reconstruction performance across a broad range of event complexities, and provide an empirical bound on failure modes for the signal classes considered in this work.

Finally, although the concept is highly promising, the engineering scalability to multi-camera, synchronized large detectors is only discussed at a conceptual level and not yet supported by experimental demonstrations of timing synchronization, mechanical tolerances, or long-term operational stability.

We agree that all these items are important for the scalability to large-scale detectors. The present manuscript, however, is intended as demonstration of the PLATON building block and of its physics potential, rather than a full engineering validation of a final, integrated multi-camera system. A comprehensive treatment of these engineering aspects would be more appropriate for a dedicated technical publication preceding detector construction.

In fact, this is the first time that a plenoptic system instrumented with SPAD arrays is built and validated with an optimised image post-processing method. Moreover, SPAD arrays are still at the engineering stage, as the technology has been proven in independent studies in literature to fulfill the requirements that we listed for particle detection.

That being said, readout electronics with timing synchronisation at the level of $O(100 \text{ ps})$ is quite standard for plastic scintillator detectors in particle physics experiments based, for example, on SiPMs, ASICs for the signal digitization and FPGAs for the time synchronisation. Another example are modern TOF-PET scanners.

Thus, it is reasonable to assume that the same performance can be obtained also with systems of SPAD arrays. This is exactly what we are developing right now.

The same can be said for the long-term operational stability of similar systems.

Concerning mechanical tolerances in the current prototype, the achieved spatial resolutions are reproducible in our optical simulation, which supports that the opto-mechanical configuration used in the measurements is consistent with the model employed for reconstruction and network training.

Alignment between different cameras can only be studied when a system made of multiple cameras is available. Thus, it is not possible to verify it in our laboratory, yet. In particle detectors, such misalignments are typically quantified and corrected using calibration data (e.g., cosmic rays, radioactive sources, lasers, LEDs), with consistent treatments in both simulation and reconstruction. The calibration scheme used for the present prototype can be extended to multiple cameras operating simultaneously, following standard procedures used in experiments that require $O(100 \text{ }\mu\text{m})$ spatial precision.

In summary, while these detailed engineering validations are beyond the scope of the present work, they represent a clear and standard programme for a future multi-camera PLATON demonstrator.

For these reasons, the manuscript requires a Major Revision to strengthen the experimental evidence, improve the realism of the simulations, and clarify the robustness of the reconstruction framework. Importantly, these issues are addressable with additional analysis and validation and do not undermine the originality or fundamental validity of the approach. In summary, the work is well suited for Nature Communications, but it requires a Major Revision to reach the level of experimental and methodological maturity expected for publication.

Reviewer #2 (Remarks to the Author):

Several shortcomings are visible in the paper, mostly related to details and context. Not much information is given on scintillator properties, average light yield (apart from being "sparse"), light yields along the path, attenuation effects and homogeneity for large scintillator volumes, scaling effects for large volumes (admittedly a technical detail), photon propagation times, potential effects from reflection and scattering inside a less perfect scintillator etc. This lack of detail makes it hard to evaluate the presented use case.

Details about the scintillator used in the prototype measurements (see Sec. 2.2.1 - L348ff) as well as in the optical simulation study (see Sec. 4.1) are reported in the text.

For the simulation, we decided to use the parameters of EJ-262 as it's the same scintillator used for the prototype measurements. On the other hand, several types of organic scintillator (both liquid and plastic) are commercially available. For instance, liquid scintillators have a typical light yield around 10,000 photons / MeV, attenuation lengths equal or above 20 m, as well as very high purity, thus, very uniform optical performance. Liquid scintillators are used in giant neutrino detectors and would be the ideal option for a final bigger PLATON detector.

For scaling to much larger volumes, the liquid scintillator optical properties are not a major concern. We are aware of the fact that the plenoptic parameters will have to be adjusted consistently. Text has been added in Sec. 4.1 (L750ff) to clarify it.

Furthermore, to give a better understanding of the optical effects, we have added details about the Geant4 setting adopted for the simulation of the optical surface and material interfaces to Sec. 4.1, L758ff. We have also added the average light yield observed in the neutrino simulations to section 2.3 (L457ff) and for point source of 1MeV in Sec 2.3.1 in L469ff and L475ff, as well as the speed of photons in scintillator and in the optical system in Sec. 4.1 (L785ff).

Concerning optical imperfections, we agree that this could be a valid concern for the actual implementation of plastic scintillator detectors. On the other hand, we can confirm from new measurements in plastic scintillator, after the submission of this paper, that we do not observe a concerning degradation of the spatial resolution (still within 500 um for light intensity lines). These measurements were obtained by exciting a EJ-262 plastic scintillator block with 2-photon absorption, that allows to reproduce the absorption/emission chain of the scintillation process. We plan to publish this work in another independent paper that we have just started to write.

Said that, one could think of a slightly different detector configuration, that consists of photocaleras inside the vessel filled with liquid scintillator and far enough (e.g. 0.5 m) from the vessel surface to avoid issues related to optical imperfections in the surfaces

Also it is noted that other general detector parameters like the overall detection efficiency (as opposed to the presented reconstruction efficiency) or the achieved vertex resolution (presumably better for events with more tracks in the same origin) are missing.

The overall detection efficiency belongs to the scintillation light output, which is around 10,000 photons / MeV (8,700 for EJ-262). This makes the pure signal neutrino event detection efficiency basically 100%, even with the PDE of SwissSPAD2 sensor used in the prototype (~5%) with a coverage of 33% of the surface (2 out of 6 faces, as simulated in the neutrino study).

If in the definition of efficiency we also include the requirement that the reconstructed position shall be within a certain distance from the actual origin of the scintillation emission, numbers can be found in the legend of Figure 8 (left plot). The failure rate in reconstructing a point-like emission source with either 2 or 8 plenoptic cameras in the scintillator volume covered by the total depth of field and field of view is zero. 68% of the events are reconstructed within less than 1mm from the true origin in any plenoptic configuration. 95% of the reconstructed events are within ~0.7 mm (~7 mm) for 8 (2) plenoptic cameras.

Finally, the proton track reconstruction efficiency is shown in Figure 5. It is higher than 90% with a threshold around 200 MeV/c.

An outer tracking detector for the produced muon is assumed, but no parameters are given. It would be worth discussing how the parameters of this detector (which presumably has to match with a track inside the scintillator volume) influence the results and would be taken into account for the application in CC neutrino interaction studied in this paper.

In our simulation, the muon position is assumed to be measured by an outer tracking detector with a resolution of 1mm and the angular resolution with 0.05 radians (see L508ff). Such resolution (or better) can be easily achieved by a time projection chamber (TPC) like the one built for the magnetized near detector (ND280) of the T2K experiment and currently taking neutrino data. Some of the authors played a central role in the upgrade of ND280 [arXiv:2511.18650, Nuclear Instruments and Methods in Physics Research A 637 (2011) 25–46] and are involved in the data analysis. Based on our experience, we believe that the determination of the muon TPC-entrance position with the simulated resolution is realistic.

Again, it is assumed that the beam parameters for the simulation are taken from T2K, but more details would be helpful here, as would be a comparison of Monte Carlo truth with the reconstruction in terms of very fundamental tracking parameters (vertex, momentum, angular resolution).

The T2K collaboration shares publicly the files of the accelerator neutrino flux used in full simulation of the experiments for the data analysis in neutrino cross section measurements at the ND280 near detector as well as at the Super-Kamiokande far detector. The following plots show the public neutrino flux used in our work and it is the one used in the search for CP violation at T2K by [e.g. PHYSICAL REVIEW D 91, 072010 (2015)]:

Since the article is already quite long, we thought to not add these plots. Given the T2K neutrino flux shown above, the NEUT generator software, exactly the same as used in T2K, simulates the neutrino interaction in the scintillator providing the list of “final-state” particles (neutrino interaction vertex, particle types and momentum vectors). This defines the truth information regarding the neutrino interaction vertex position, with respect to which the vertex resolution quoted in Sec. 2.3.2 L543ff is computed. The choice of simulating all the neutrinos along the same direction is justified by the wide angular distribution that corresponds to a transverse range of more than 10 meters, as reported in [Phys.Rev.D 87 (2013) 1, 012001]:

Then, the propagation of the final-state particles in matter is simulated using Geant4. This is briefly described in Section 2.3.2 L523, with details added in Section 4.1

Other detector technologies are being developed addressing the same problem, notably large scale liquid Argon TPC and a more niche application using opaque scintillators. None of these are discussed in the paper, while a comparison especially with the former would certainly be worthwhile (if only to put the performance in context).

Thanks for the suggestion. We agree that both liquid argon TPCs and opaque scintillators are relevant and should be mentioned. Thus, we added a paragraph in the introduction in Section 1 (L26ff) about opaque scintillator and another one in the discussions in Section 3 (L710ff) about LArTPC.

Last not least, it is easy to conceive that this particular method will have interesting applications beyond high energy neutrino physics. The paper would benefit from an outlook into the wider applications of this ingenious technology.

The PLATON concept is suitable for positron-emission tomography (PET), for which we recently submitted three patent applications, neutron radiography, muon tomography, synthetic computed tomography (CT), and proton CT. This list is provided at the end of Section 3.

Reviewer #3 (Remarks to the Author):

ll 30-34) The authors point out the necessity of large numbers of read-out channels as a draw-back of existing detector technologies, implying that this new technology will be superior in that regard. But if the detector walls are covered in plenoptic cameras, with 10s of thousands of single addressable pixels each, doesn't that also mean an incredibly large number of "channels"?

We thank the reviewer for highlighting the need for more context about the number of channels in different technologies, as it can be used in a slightly different way in traditional SiPM-based instrumentation compared to that of SPAD arrays. In the text, we now have clarified that in SiPM systems a channel typically is a full analog chain, requiring isolated signal lines, dedicated digitisation electronics and controllers for data packaging. From each channel one obtains the number of counts (signal photons or noise) recorded by a single SiPM with a typical active surface of $O(1\text{mm}^2)$.

In the SPAD array sensors deployed to collect sparse-photon images, each pixel is equivalent to an independent SiPM, but with a much higher spatial resolution. The SPAD arrays discussed in this article also integrate the digitization electronics directly on-chip. This allows millions of individual pixels to be read out via a shared digital data line (e.g. a single 4Mbit package for 4 million pixels), significantly reducing the required external electronics. Here each such shared data line can be considered as a single read out channel. We have added some clarification text in Section 1, L 42ff and L58ff.

ll 293-302) This paragraph is impossible to understand when reading the paper for the first time. What does "augmented with additional data points" mean? I think this paragraph should either explain the "likelihood" method in more details, or in fewer. Just saying, that the authors tested an alternative method based on the likelihood of the seen light patterns and then quoting the achieved resolutions should be enough here.

We have adjusted the description of likelihood method to be more concise in the results section (Sec. 2.2 L320ff), and moved any further explanation to Methods (Sec. 4.4 L1038ff).

ll 339-342) The authors mention that the prototype had the scintillator cube "out of focus". But they fail to mention what that means for the reconstruction. Does it even matter, since the path of every single photon is reconstructed individually? Why was it not possible to re-focus the objective on the cube?

For a plenoptic camera exposed to intense light sources, when operated in focus, a single point source would lead to a single pixel below the corresponding microlenses. When the camera is operated out of focus, the images of the microlenses contain a focus-blur, meaning that more than 1 pixel is active. In the image reconstruction this introduces a higher level of ambiguity among different pixels, thus to a decreased accuracy in the prediction of the arriving photon angle. In simulations we have seen that

the expected focus blur is ~ 2 pixels in radius. While this affects the resolution with which the angle of the arriving photon can be detected, in simulations we have found, that for the limited photon regime, this is a secondary impact on the 3D reconstruction resolution, which is dominated by the low number of photons detected. This was addressed in sec 4.2, L860ff.

Moreover, the refocusing would have required a new optical calibration of the main lens parameters. Doing so would have not been very much in the spirit of a 3D camera.

We have added short clarifications to the results section L356ff and L388ff, Sec 2.2.1.

Fig 3) It looks like even with the "maximised light collection" the authors have only recorded a handful of events with 4 photons. Is this consistent with the expected light yield when considering solid angle coverage and sensor efficiency?

What is the total "exposure time" to collect these few events?

I.e. how many beta rays were not detected, or are indistinguishable from the noise?

The overall capture time is roughly 100s, using approximately 10.7M frames, with an exposure time per frame of 10 μ s.

Yes, the number of signal counts that we obtain are consistent with our expectation based on the parameters of the system during the measurement. Due to the low number of photons produced and captured, only events in the high-energy tail of the 90Sr spectrum will produce enough light to be distinguished from the background distribution. Even in this case, in simulations we expect only up to 3 photons. The majority of Sr90 events captured will not produce enough light to be seen.

It is not possible to tag electrons event by event, unless we constrain our selection only to those events with more than 3 counts. In fact, the bkg-only sample does not contain any event with that number of counts. Also, we performed a bkg-only simulation with dark counts randomly sampled from our SwissSPAD2 data. We simulated a number of frames x100 higher than that of the 90Sr data sample. The mean dark count rate corresponding to the 90Sr sample statistics is 0.044 counts per frame. From a toy study, after normalizing to the same number of frames as in the 90Sr data sample, we obtain only around 0.17 frames that contain 4 or more counts and fall in the scintillator region within 20mm far from the true position of the 90Sr source. This has been addressed in L401ff, Sec 2.2.1.

d) Is this the 3D residual? The authors went through the trouble of defining all sorts of distances in Eq 1-3. Why not use these defined names in the plots? Instead, everything is called "distance", "residual", or even more confusingly "true - reco" in different plots.

Yes, that's correct. The figures have been updated accordingly.

I 437) It is somewhat confusing that every single detector is called "PLATON", while there are three distinct setups discussed in the paper: the single-camera prototype, the eight-camera simulation, and the 1 m³ simulation. These should be given distinct names, so it is clear what is being talked about in each section.

That's a good suggestion. We changed in the manuscript the names of the different prototype or simulated configurations into: PLATON-prototype, PLATON-10cm and PLATON-1m

Fig 4c) If the cameras are focussed on the cube, why are the collected images so smeared out? It's probably not the MLA, since the microlenses should be very small on the scale of the picture. Is it because the depth of focus is so narrow?

What appears here to be a smeared image is what is expected for a plenoptic camera in focus. We have added a clarifying statement to Sec 4.2 L859ff.

The definition of "in focus" is also discussed in our reply to the 3rd comment. It does not refer to the main lens but to the number of pixels activated under a single microlens when the plenoptic camera is exposed to a single-point light source.

In this figure, the blurred image appears to be the ensemble of sub-images formed under different microlenses, each with a diameter of ~ 1 mm in this camera.

When the camera images a single straight particle track, its path is projected as multiple segments, each one in a different microlens. The number of microlenses that capture the segment defines the "Virtual Depth". This value is directly proportional to the physical distance of the object from the main lens, allowing for depth estimation.

d & e) The y axis seems to be some squared distance between true and reconstructed position. Is it 2D or 3D? Looks like the square of some L^2 norm. But if it is squared, why is the unit "mm" and not "mm²"?

It is the 3D distance. We have fixed the typo and updated the plots accordingly.

Fig 5g) Is this the momentum of the leading proton or of both? What about the 3rd and 4th protons in the "3p" and "4p" contamination? How can there be a "1p" contribution? How is the purity defined? Usually, a purity as function of true quantity makes no sense. Is this just the ratio of "2p" in this plot? The y axis label is a bit confusing, as it could be read as "ratio of efficiency to purity". Should the horizontal lines be part of the legend? Are those numbers somehow significant? Or would it be better to just have horizontal grid line in the plot.

We clarify that the x-axis represents the true momentum of the leading (highest momentum) proton in the event. Regarding the 1p contribution (light blue): the plot shows the composition of the selected sample; these are events with a true $1p_{1\mu}$ topology that were incorrectly identified as $2p_{1\mu}$ by the selection algorithm. Since they contain one real proton, they are plotted according to that proton's true momentum. We have clarified it in the caption of the Figure.

Yes, the purity curve is mathematically identical to the ratio of the signal component (2p) to the total bar height in each bin. We overlay the curve explicitly so the reader can see the quantitative trend of the contamination without having to visually estimate ratios from the stacked histogram.

We have updated Fig. 5 in the manuscript to make sure it is more understandable now.

Fig 6) I think it would be nice to mention how many cameras and how many pixels in total this setup has.

The tonne-scale configuration in Fig. 6 employs 800 plenoptic cameras, each with about 4M pixels per camera, for a total of ~3.2 billion pixels. We have added this information to the caption of Fig. 6.

ll 654-656) Could the authors substantiate the claim that this technology is well-suited for neutrinoless double-beta decay? It seems to me like the most important aspect for that is the energy resolution, and it was not shown that PLATON is superior in that regard.

Our intent in this paragraph is to show neutrinoless double-beta decay experiments as an application where PLATON could improve the state of the art, potentially. We agree that the paragraph might seem too strong, and we weakened the claim in the text by mentioning PLATON as a potential technology that might deserve future studies in this context. We also updated the list of articles cited in the text.

We agree that the smoking gun for neutrinoless double-beta decay experiments is the energy resolution, especially for rejecting the irreducible background from double-beta two-neutrino decay. On the other hand, to our understanding, further background rejection would be achieved if particle tracking is also enabled. An example is the identification of Compton scattering events and single-beta decays.

Although we defer confirmation of the suitability of the PLATON technology for further studies, it might improve on some existing neutrinoless double-beta decay technologies based on scintillator detectors.

The first example that we cite is the SuperNEMO experiment. Here, the calorimeter module is very similar to the PLATON-10cm design, but instead of 3D imaging, it simply adopts a single PMT. The advantage of a 3D imaging system would be the possibility to identify two Bragg peaks from the emitted electrons with a more accurate determination of background topologies such as single electrons, positrons, Compton scattering or alpha's.

At larger scales, examples might be large liquid scintillator detectors similar to KamLAND-Zen or SNO+, loaded with nuclei optimal for neutrino-less 2beta decay searches (e.g. Tellurium, Xenon). Although we have not studied the technological limit of PLATON (e.g. smaller pixels, optimisation of the optics for large volumes), it might maximise the fiducial volume of these large-scale detectors, important for low-rate experiments, and, possibly, improve the topological identification of the different background sources.

ll 658-664) Could the authors substantiate the claim that PLATON will be well suited for large scale detectors and improve the Cherenkov ring reconstruction? Covering a kiloton scale detector with plenoptic cameras seems like a quite expensive proposal.

We made an educated guess for a very large-scale production (>100,000 wafers), equivalent to 10,000 20-inch PMTs (e.g., Super-Kamiokande using the newer Hyper-Kamiokande PMTs). With a reference cost of 2.5 kCHF per 8-inch wafer, given the entity of such very large mass production, the total cost could be further reduced. Thus, we assume a scaling factor between 2 and 4. The result is that the cost of 20-inches of SPAD sensors ranges between a factor of 2 and 4 higher than that of a 20-inch PMT (assuming ~3 kCHF per PMT, already extrapolated to very big mass production). We expect the optics (microlens array and main lens) to be not a driving cost. It might be even possible to obtain an even larger reduction of the wafer cost but this will have to be checked.

Moreover, it's worth saying that the dominant fraction of the costs of these very large-scale experiments is the excavation and the preparation of the cavern. Being many of such facilities already available in several countries (e.g. Super-K, Hyper-K, JUNO, SNO), in the future experiments one might consider to invest more resources in the technology (such as PLATON, if proven to be the right one) rather than in the facility itself.

Concerning the improvement of the water Cherenkov ring resolution, the capability of PLATON in inferring the direction of the Cherenkov photons independently from the actual detection of the ring would introduce additional information not available in state-of-the-art detectors that can only rely on the directionality of the Cherenkov light.

The issue is that the detected ring is a superposition of multiple rings as a result of, for example, the primary electron scattering processing, EM showers or the energy loss, e.g. by the muon, that causes the ring to close when reaching the Cherenkov radiation threshold. PLATON might allow to achieve a Cherenkov ring sub-resolution, by combining the known directionality of Cherenkov radiation with the PLATON's capability of inferring the photon angle.

II 675-680) Could the authors substantiate their claim the PLATON has "enormous" potential in these applications? What makes it potentially better than existing solutions?

By "enormous potential" we refer to the combination of features demonstrated or projected to 1m³ size in this work: sub-millimetre 3D spatial resolution for 1 MeV energy deposition in a 10×10×10 cm³ scintillator module, few-millimetre single-point resolution at the tonne scale with a path to sub-millimetre performance, and single-photon, sub-nanosecond timing in an unsegmented volume. These figures are already competitive with, or better than, state-of-the-art organic scintillator detectors. Since spatial resolution, depth-of-interaction capability and detection efficiency are key limitations of current PET, pCT and fast-neutron imaging systems, this combination suggests that PLATON could offer substantially improved performance in these applications, which motivates our statement. We have clarified this point in the text and softened the wording.

II 821-823) By how much did the glue reduce the field of view? Did that matter at all for the presented results?

The angular field of view shrunk from 19 deg to roughly 13 deg. In the results presented, the main impact is the narrower field of view (FoV) on the calibration measurements. This can be seen in Fig 2b, where the front lateral rows of points do not contain the same number of test-points, due to the decreased FoV.

Eq 4) Why is the sum divided by the number of pixels? Just the sum alone gives the log likelihood of the given pattern. Why divide that by the number of pixels? It should not matter for the mere finding of a minimum though.

We thank the reviewer for the remark. We use the binary cross-entropy in its standard “average per pixel” form from the machine-learning community, hence the division by the number of pixels (N_{pixels}). As the reviewer correctly points out, this overall normalisation does not affect the position of the minimum with respect to \vec{y} ; it only rescales the loss so that its value remains $O(1)$ and independent of the number of pixels used (e.g. when changing the region of interest - ROI - or masking pixels). We have clarified this after Eq. (4)

ll 885 & 927) Why is this called “fake data”? What makes this “fake” as opposed to any other simulations done in this work?

By “fake data” we meant data-driven pseudo-data built from the measured calibration images: after subtracting the independently measured dark-count map, we treat each intensity image as an empirical PDF and generate photon-starved binary frames by random sampling from it. In contrast to the Monte Carlo samples used elsewhere, these frames do not rely on the Geant4-based optical simulation but inherit the full optical response of the prototype, with only the photon statistics resampled. To avoid confusion, we now replace “fake data” with “sampled data” / “data-driven pseudo-data” and clarify this point in Sec. 4.4. (L996ff).

ll 1087-1079) What is this “regularisation” mentioned here? Why is it needed? What happens if it is omitted?

In this context, “regularisation” refers to the stochastic sub-sampling of photons when an event contains fewer tokens than the maximum sequence length. For such events we randomly sample 90% of the photons at each epoch; as a result, the same physical event is seen with different photon combinations over training. This acts as an input-level regularisation/data-augmentation mechanism, reducing overfitting to specific photon patterns and improving generalisation to unseen events. Regularisation is a standard term in machine learning. We have rephrased the paragraph (L1189ff) in the manuscript to better explain the mechanism.

REVIEWER COMMENTS

Reviewer #1 (Remarks to the Author):

No requests.

Reviewer #2 (Remarks to the Author):

No requests.

Reviewer #3 (Remarks to the Author):

Fig 4d,4e,5b (maybe 8b, 8d), 8e, 8f are still not using the distances defined in eq. 1-3.

- We kept a uniform distance label in these figure axes to make the figures more self-explanatory and consistent with journal guidelines.

In the answer to our comment regarding Fig 3, the authors answered that the number of events from the ^{90}Sr source is consistent with the expectation, but they don't say how many events with more than 4 photons are expected from the source. (It is however clear that it cannot be explained with the background only.)

- In 1,000 toy Monte Carlo runs, the number of events with 4 or more counts ranges from 6 to 12 for the corresponding number of simulated frames.

Reviewer #4 (Remarks to the Author):

No requests.